

# Understanding the oxidants transition and SOA property in limonene ozonolysis: Role of different double bonds, radical chemistry, and water

Yiwei, Gong[1], Zhongming Chen[1], and Huan Li[1]

[1]State Key Laboratory of Environmental Simulation and Pollution Control,

College of Environmental Sciences and Engineering, Peking University, Beijing 100871, China

*Correspondence to*: Z. M. Chen (zmchen@pku.edu.cn)

**Abstract.** Volatile organic compounds (VOCs) play an important role in air quality and climate change, largely because of their contribution to atmospheric oxidation capacity and secondary organic aerosol (SOA) formation through their oxidation. In this study, a series of products including peroxides and carbonyl compounds in both gaseous and particulate phases were simultaneously detected to help us investigate the oxidants transition and SOA property in limonene ozonolysis. Reactants ratio, OH radical scavenger and relative humidity (RH) were controlled to discuss the effect of endocyclic and exocyclic double bonds (DBs), radial chemistry and water. Alkene ozonolysis not only consumed but also regenerated oxidants, which made a great impact on atmospheric chemical processes. For this issue, we first paid attention to the generation of stabilized Criegee intermediates (SCIs) and OH radical. The variation of $H_2O_2$ and hydroxymethyl hydroperoxide (HMHP) formation with RH showed the importance of the reaction with water for limonene SCIs, and the estimated SCIs yields of endocyclic and exocyclic DBs were ~ 0.24 and ~ 0.43, respectively. OH yield was determined by adding sufficient OH scavenger, and the OH yields of endocyclic and exocyclic DBs were ~ 0.65 and ~ 0.24, respectively. The results indicated that in limonene ozonolysis the endocyclic DB was inclined to generate OH radical through hydroperoxide channel, while the exocyclic DB had higher fraction of forming SCIs. Besides,

other gas-phase and particle-phase peroxides were also studied. The formation of peroxyformic acid (PFA) and peroxyacetic acid (PAA) were promoted significantly by the increasing RH and the oxidation degree, and the discrepancy between the experimental and model results suggested some missing formation pathways.

Considerable $H_2O_2$ generation from SOA in aqueous phase was observed especially at high $[O_3]/[limonene]$, which was mainly attributed to the hydration and decomposition of particulate unstable peroxides such as peroxycarboxylic acids and peroxyhemiacetals. Different DBs and radical chemistry revealed their influence on aerosol property through affecting the behavior of SOA on generating $H_2O_2$. As a species owning high SOA formation potential, another key issue we investigated in limonene ozonolysis was SOA property, for which

peroxides and carbonyls were chosen as representatives. The results showed that in limonene SOA, peroxides could account for 0.07–0.19 at low $[O_3]/[limonene]$ and 0.40–0.58 at high $[O_3]/[limonene]$, which confirmed the important contribution of peroxides to aerosol formation. The partitioning behavior of peroxides showed that multigeneration oxidation helped produce more low-volatility peroxides, which provided some explanation for higher SOA yield. The partitioning behavior of carbonyls was also discussed and the experimental partitioning

coefficients ($K_p$) were usually several orders of magnitude higher than theoretical values, yet the relationship of $K_p$ observed in laboratory with vapor pressure offered some reference for predicting the contribution of carbonyls to SOA formation. This study provided new insights into the oxidants transition and SOA property in limonene ozonolysis, and limonene showed its specificity in many aspects when both endocyclic and exocyclic DBs were ozonated. We suggested that the atmospheric implications of terpenes containing more than one DB

and the properties of particulate products especially peroxides still needed further study.

## 1 Introduction

As an important monoterpene, limonene has a high emission rate both from biogenic and anthropogenic sources,



which is only second to pinene (Atkinson and Arey, 2003; Clausen et al., 2001; Fellin and Otson, 1994; Griffin et al., 1999; Guenther et al., 1995; Lamb et al., 1993; Seifert et al., 1989; Sindelarova et al., 2014; Wolkoff et al.,

2000). Total monoterpene emission amount was estimated to be 50 Tg C yr$^{-1}$, and limonene might comprise about 20% of that (Stroud et al., 2005). In addition to the large quantity of emission from vegetation, its extensive utilization in household and industrial processes also makes limonene unnegligible in the atmosphere. Compared with its isomer α-pinene and β-pinene, an obvious feature of limonene in structure is that it owns two different double bonds (DBs): an endocyclic one and an exocyclic one, which makes the behavior and the fate

of limonene in atmosphere complicated. Terpene oxidation is a well-known source of secondary organic aerosol (SOA), and limonene has proven to have higher potential of SOA formation than α-pinene because it is doubly unsaturated. The considerable potential of producing low-volatility compounds makes limonene have an important contribution to SOA formation (Andersson-Skӧld and Simpson 2001; Kroll and Seinfeld, 2008; Lane et al., 2008; Lee et al., 2006). Some studies have investigated the initial process and SOA yield of limonene

oxidation (Grosjean et al., 1992, 1993; Glasius et al., 2000; Leungsakul et al., 2005; Pathak et al., 2012), however, the knowledge of detailed reaction mechanism and SOA property in limonene ozonolysis remains unclear, especially when it comes to the effect of different DBs, which deserves efforts of further study to help us obtain a better understanding of limonene chemistry and its implications in the atmosphere.

Reactions of alkene with ozone have been explored by numerous researches because of their role as important

sources of free radicals, intermediate products, and aerosol. The first step of alkene ozonolysis is the addition of O$_3$ to the carbon-carbon double bond forming an energy-rich primary ozonide (POZ), which decomposes to two sets of carbonyls plus carbonyl oxides called excited Criegee intermediates (ECIs) (Criegee, 1975; Fenske et al., 2000; Gutbrod et al., 1996, 1997; Kroll et al., 2001a, b). ECIs can isomerize through hydroperoxide channel followed by OH production, rearrange to eaters with subsequent decomposition, or undergo collisional



stabilization forming stabilized Criegee intermediates (SCIs) (Cremer et al., 1993; Presto and Donahue, 2004;

Richard et al., 1999; Wegener et al., 2007). Some products formed from alkene ozonolysis that own the power

of removing oxidizable compounds play an extremely important role in atmospheric chemical processes and

evolution. It means that alkene ozonolysis is not only a process of consuming oxidants, but also a process of

regenerating oxidants. These species contribute much to the atmospheric oxidation capacity and control the

atmospheric self-cleaning process by removing a series of trace gases (Möller, 2009; Prinn, 2003; Taatjes et al.,

2013). In recent years the reactive oxygen species (ROS), which includes oxygen-related free radicals (e.g., OH,

$HO_2$, and $RO_2$), ions and molecules (e.g., $H_2O_2$, organic and inorganic peroxides), attracts increasing attention

because of its relation with atmospheric oxidative level and adverse health effect caused by particle-phase ROS

(Gallimore et al., 2017; Huang et al., 2016; Wragg et al., 2016). Alkene ozonolysis is thought to be an important

source of OH radical in the atmosphere, which makes OH-initiated reactions possible to continue in the dark

(Kroll et al., 2001a, b). SCIs generated during the process prove to have sufficient lifetime to react with other

trace species, such as $H_2O$, $SO_2$, $NO_2$, carbonyls, alcohols, carboxylic acids, etc., enhancing the atmospheric

oxidation capacity and promoting the SOA formation (Sakamoto et al., 2017; Sipilä et al. 2014; Yao et al.,

2014). Furthermore, reactions of alkene with ozone can also generate considerable amount of peroxides, which

receive much attention due to their oxidizability and role as radicals reservoir. $H_2O_2$ is the most crucial oxidant

to oxidize S (IV) forming sulfuric acid and sulfate in the aqueous phase, and organic peroxides could also

oxidize some species (Calvert et al., 1985; Penkett et al., 1979; Peña et al., 2001). Overall, although alkene

ozonolysis consumes ozone, these critical compounds formed from the reaction contribute much to atmospheric

oxidizing capacity. The existing knowledge of oxidants transition in alkene ozonolysis is not sufficient, yet it is

essential for us to evaluate the actual impact of this reaction on atmospheric evolution and human health.

One important reason for limonene chemistry drawing much attention is its high SOA formation potential.



Although progress has been made over the past years on simulating SOA formation with the theory of gas-particle partitioning, there are still large discrepancies between the model and experimental results (Cocker et al., 2001; Griffin et al., 1999; Hoffmann et al., 1997; Odum et al., 1996; Pankow, 1994; Presto et al., 2005;

Pye and Seinfeld, 2010). Laboratory studies about SOA formation in limonene ozonolysis mainly focused on the aerosol yields under different conditions and identifying some products in the particulate phase (Calogirou et al., 1999; Leungsakul et al., 2005; Ng et al., 2006), however, the composition and property of limonene SOA still need detailed study. As a double-unsaturated terpene, SOA formation process of limonene could be more complicated than single-unsaturated terpene, as the multigeneration oxidation has significant influence on SOA.

In this study, two classes of species: peroxides and carbonyls are chosen to study their behaviors in limonene SOA formation. In the last few years, organic peroxides have been analyzed and suggested to be an important composition in aerosol (Docherty et al., 2005; Heaton et al., 2007; Li et al., 2016; Mertes et al., 2012; Pathak et al., 2012), and particulate peroxides could cause negative health effect after penetrating into lungs (Verma et al., 2009; Wragg et al., 2016). The reactive uptake and particle-phase reactions of carbonyls are believed to be

responsible for fractions of aqueous SOA formation (Ervens et al., 2011; Mcneill et al., 2012), especially for dicarbonyls glyoxal and methylglyoxal. Up to now, few researches pay attention to the contribution of peroxides and carbonyls to limonene SOA and their behaviors in aerosol formation are vague in our knowledge.

Other key issues we are concerned about are the water and radical chemistry effect. Due to the sufficient water vapor in the atmosphere, it makes a great difference on several chemical and physical processes. It is necessary

to design and conduct experiments in a wide range of relative humidity (RH) to provide more valuable information when the results obtained in laboratory are applied to the actual circumstances. Researches before have provided evidence for OH radical formation in ozonolysis experiments and OH scavenger is often used to avoid the disturbance of OH reaction. However, it should be noted that when OH scavenger removes OH radical

from the reactor, it could bring other reactions into the system, which would affect the production of

hydroperoxy ($HO_2$) and alkylperoxy ($RO_2$) radicals. Some studies suggested that the choice of OH scavenger

influenced the $HO_2/RO_2$ and SOA yield (Docherty and Ziemann, 2010; Keywood et al., 2004), yet its effect on

other products was seldom discussed. It was proposed that to simulate the real atmospheric environment,

experiments should be conducted at high $HO_2/RO_2$ and low reactant concentration (Jonsson et al., 2008a). Chew

and Atkinson (1996) argued that the ability of 2-butanol and cyclohexane in scavenging OH radical was similar,

however, 2-butanol was proven to give more $HO_2$ radical than cyclohexane, suggesting that 2-butanol was a

more appropriate scavenger. Here, to study the impact of water and radical chemistry on the reaction system, a

series of experiments were preformed under various RH conditions in the presence or absence of different OH

scavengers (2-butanol or cyclohexane).

The focus of this study is to investigate the oxidants transition and SOA property in limonene ozonolysis,

especially for the role of different DBs, radical chemistry, and water. On the one hand, through determining the

generation of SCIs, OH radical, and peroxides, the issue of oxidants transition in the reaction system is

discussed. On the other hand, peroxides and carbonyls are taken as representative to study their behaviors in

SOA formation. Reactants ratio, OH scavenger, and RH are controlled to explore the effect of different DBs,

radical chemistry, and water.

**2 Experimental**

**2.1 Chemicals**

R-(+)-Limonene (Sigma-Aldrich, ≥ 99.0%), 2-butanol (Sigma-Aldrich, 99.5%), cyclohexane (Alfa Aesar, ≥

99.9%), potassium iodide (KI, Alfa Aesar, 99.9%), hydrogen peroxide ($H_2O_2$, Alfa Aesar, 35wt.%),



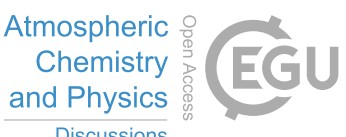
ortho-phosphoric acid ($H_3PO_4$, Fluka, 85–90%), hemin (Sigma, $\geq$ 98.0%), 4-hydroxyphenylacetic acid (Alfa

Aesar, 99%), formaldehyde (Sigma-Aldrich, 37wt.%), acetaldehyde (Amethyst Chemicals, 40wt.%), acetone

(Fluka, $\geq$ 99.7%), hydroxyacetone (Sigma-Aldrich, 90%), glyoxal (Sigma-Aldrich, 40wt.%), methylglyoxal

(Sigma-Aldrich, 40wt.%), 2-butanone (Alfa Aesar, $\geq$ 99%), acetonitrile (Alfa Aesar, $\geq$ 99.7%), 2,4-dinitrophenyl

hydrazinel (DNPH, TCI, 50wt.%), ammonia solution ($NH_3 \cdot H_2O$, Beijing Tongguang Fine Chemicals Company,

25.0–28.0%), ammonium chloride ($NH_4Cl$, Beijing Tongguang Fine Chemicals Company, $\geq$ 99.5%), sulfuric

acid ($H_2SO_4$, Xilong Chemical Company, 95.0–98.0%), ultrapure water (18M$\Omega$, Millipore), $N_2$ ($\geq$ 99.999%,

Beijing Haikeyuanchuang Practical Gas Company Limited, Beijing, China), $O_2$ ($\geq$ 99.999%, Beijing

Haikeyuanchuang Practical Gas Company Limited, Beijing, China), polytetrafluoroethylene (PTFE) filter

membrane (Whatman Inc., 47mm in diameter), and quartz microfiber filters (Whatman Inc.) were used in this

study.

**2.2 Apparatus and procedures**

A flow tube reactor (2 m length, 70 mm inner diameter, quartz wall) equipped with a water jacket for controlling

temperature was used to investigate the ozonolysis of limonene. All the experiments were conducted at 298±0.5

K in the dark. $O_3$ was generated by $O_2$ photolysis in a 2 L quartz tube with a low-pressure Hg lamp, and the

detailed quantification method of $O_3$ was described in our previous study (Chen et al., 2008). $H_2O_2$ produced by

UV irradiation of $O_2$ and trace water was measured in control experiments and deducted from the results.

Limonene gas was generated by passing $N_2$ flow over liquid limonene in a diffusion tube at the selected

temperature and OH scavenger (2-butanol or cyclohexane) gas was generated with a bubbler. The

concentrations of limonene and OH scavenger were determined by gas chromatography with flame ionization

detector (GC-FID, Agilent 7890A, USA). Water vapor was generated by passing $N_2$ through a water bubbler,



which contained a carborundum disc submerging in ultrapure water (18 MΩ). The mixing gas including limonene, OH scavenger, ozone, and dry or wet synthetic air (80% $N_2$ and 20% $O_2$), was successively introduced into the reactor, and with a total flow rate of 2 standard L min$^{-1}$, the residence time was estimated to be 240 s.

To explore the reaction mechanism of endocyclic and exocyclic DBs ozonolysis, and the effect of

multigeneration oxidation in limonene ozonolysis, two sets of experiments with different ratios of ozone to limonene concentration were conducted. In the following content, [$O_3$] denoted the concentration of ozone, [limonene] denoted the concentration of limonene, and [$O_3$]/[limonene] denoted the ratio of ozone to limonene concentration. In the low [$O_3$]/[limonene] set of experiments, the initial concentrations of limonene and ozone were ~ 280 ppbv and ~ 500 ppbv, respectively. In the high [$O_3$]/[limonene] set of experiments, the initial

concentrations of limonene and ozone were ~ 183 ppbv and ~ 19 ppmv, respectively. In both sets of experiments, enough 2-butanol and cyclohexane were added to scavenge OH radical in the RH range of 0–90%. In the tables and figures, the low and high ratio sets of experiments were denoted with mark L and H, and the conditions in the absence of scavenger, in the presence of 2-butanol and in the presence of cyclohexane were represented by No-sca, 2-But, and C-hex, respectively. Experimental conditions were listed in Table 1.

According to previous studies, in limonene ozonolysis the rate constant of endocyclic DB reaction with ozone was $2 \times 10^{-16}$ cm$^3$ molecule$^{-1}$ s$^{-1}$ (Atkinson, 1990; Shu and Atkinson, 1994), while the exocyclic DB reaction with ozone was about 30 times slower than endocyclic DB (Zhang et al., 2006). Based on those rate constants we estimated that at low [$O_3$]/[limonene], less than 1% exocyclic DB was ozonated, so this situation mainly represented the firstgeneration oxidation. In this circumstance, because the ozone concentration was low, OH

reaction would impact the amount of limonene consumed by $O_3$. In the presence of OH scavenger, ~ 42% endocyclic DB reacted with $O_3$, while in the absence of scavenger, ~ 38% endocyclic DB reacted with $O_3$. At

high [O₃]/[limonene], more than 99% endocyclic DB and about 51% exocyclic DB reacted with ozone, and

since the ozone concentration in this situation was high, the OH effect on ozonolysis was presumed to be

unimportant. The latter condition, which contained multigeneration oxidation process, was more likely to

happen since the ratio of [O₃] to [limonene] was similar to the ratio in the real atmosphere.

It should be noted that one advantage of flow tube reactor was that the wall would be in equilibrium with the

gaseous phase after a stationary period, and according to our observation, this process usually needed about 2 h.

In order to stabilize the system and diminish the wall effect as much as possible, the reactor was usually aged

for 2 h prior to measurement and after experiments the reactor was rinsed out with ultrapure water and blown to

dry with N₂.

**2.3 Products analysis**

To better investigate the gas-particle partitioning of products formed in limonene ozonolysis, we analyzed

gas-phase and particle-phase products simultaneously. The formation of total peroxides and a series of

low-molecule-weight (LMW) peroxides were measured here. For particle-phase peroxides detection, a PTFE

filter was used for SOA collection and the mass of SOA was measured by semi-micro balance (Sartorius,

Germany). Since the control experiment results showed that long-time collection led to the loss of some

peroxides in particles, the collection time was controlled to be 3 h for each filter, and the accuracy of particulate

products analysis was discussed in the Supplement. Each loaded PTFE filter was extracted with 20 mL H₃PO₄

solution (pH 3.5) using a shaker (Shanghai Zhicheng ZWY 103D, China) at 180 rpm and 4 °C for 15 min, then

the SOA solution was analyzed at once. The extraction efficiency was confirmed in our previous work (Li et al.,

2016), and this method could be regarded as a reliable way to determine particulate peroxides. The peroxides

that were detected by high performance liquid chromatography (HPLC) were regarded as LMW peroxides,



while for the peroxides undetermined by HPLC, we considered them as high-molecule-weight (HMW) peroxides in the following discussion. In SOA extract solution, LMW peroxides were hardly detected, indicating that the particle-phase total peroxides concentration could be treated as the particle-phase HMW peroxides concentration. For gas-phase peroxides detection, gas through the filter was collected in a coil collector with $H_3PO_4$ solution (pH 3.5), which was detected immediately.

The detection method of peroxides was reported in our previous studies (Hua et al., 2008; Li et al., 2016), so only a brief description was given here. LMW peroxides were analyzed by HPLC (Agilent 1100, USA) coupled with post-column derivatization and fluorescence detection on line. Peroxides separated by column chromatography reacted with *p*-hydroxyphenylacetic acid (POPHA) under the catalysis of hemin forming POPHA dimers, and then the dimers were quantified by fluorescence detector. The concentration of total peroxides ($H_2O_2$, ROOH, and ROOR′) was determined by iodometric spectrophotometric method, which based on the reaction of peroxides and iodide ions (Docherty et al., 2005; Mutzel et al., 2013). Briefly, excessive KI solution was added into samples purged of $O_2$, after staying 12–24 h in the dark for derivatization, the $I_3^-$ ions produced were quantified at 420 nm by UV/VIS spectrophotometer (SHIMADZU UV-1800, Japan).

To measure particle-phase carbonyls, SOA was collected onto a quartz microfiber filter for 3 h, after collection the filter was put upside down in a conical flask. 5 mL acetonitrile, 1 mL DNPH saturated solution, and 50 μL $H_2SO_4$ solution (0.25 M) were added into the flask in sequence, the flask was then shaken at 180 rpm and 4 °C for 3 h and kept in dark for 12–24 h waiting for detection afterwards. For gas-phase carbonyls measurement, gas through the filter was directly introduced into a Horibe tube, which was placed in a cold trap (Beijing Tiandijingyi TH-95-15-G, China) at about -98 °C to freeze the products in tube. The Horibe tube was made of an inlet tube (25 cm length, 4 cm O.D.), a coil (7 laps, 1 cm O.D.), and an outlet tube with a carborundum disc. After collection, 10 mL acetonitrile was added to rinse the Horibe tube inside to dissolve carbonyl compounds,



and then the solution was mixed with DNPH saturated solution and $H_2SO_4$ solution to derivatize for 12–24 h in

the dark. Samples derived were analyzed by HPLC with UV detection (Agilent 1100, USA), and details of the

process could be found in our previous work (Wang et al., 2009).

**2.4 Wall loss experiments**

To make results more accurate, we designed and conducted a series of control experiments to quantify the wall

loss effect. Two types of experiments were carried out, including gaseous products and aerosol wall loss

evaluation. As for gaseous products, peroxides and carbonyls were chosen to analyze. Gas containing peroxide

constituents was generated by passing $N_2$ through a diffusion tube, which contained certain peroxide solution in

it. The synthetic method of multiple organic peroxides was described in our previous work (Huang et al., 2013).

Gas containing carbonyl constituents was prepared by injecting liquid substance into an evacuated steel canister

(15 L, Entech Instrument), and then $N_2$ was added constantly until the pressure in canister reached 30 psi. The

outlet of canister was linked with a mass flow controller to regulate the gas flow rate. The gas containing

peroxides or carbonyls was mixed with synthetic air at different RH and introduced into the reactor with a rate

of 2 standard L $min^{-1}$, and the concentrations of peroxides and carbonyls were controlled to be at the similar

level with the products observed in limonene ozonolysis. After the gas mixture was introduced into the flow

tube, around 2 h was needed for gas and wall to become balanced, and then the measurement would start. The

aerosol wall loss experiment used two-stage reaction equipment containing two flow tube reactors. Because we

wanted to explore the wall loss effect on pure SOA, the first reactor was used to generate aerosol where

limonene and $O_3$ had sufficient time to react completely. The gas containing aerosol out of the first reactor was

mixed with synthetic air at different RH and introduced into the second reactor with a rate of 2 standard L $min^{-1}$.

The particles at the inlet and outlet of the second reactor were collected on PTFE filters and measured by

balance to calculate the SOA concentration. The wall loss fraction of gas-phase constituent or SOA was

determined as the difference between the inlet and outlet concentration divided by the inlet concentration, which

could be expressed as ([In] - [Out]) / [In]. The wall loss experiments were conducted in the RH range of 0–90%,

and the profiles of loss fractions as a function of RH could be used for correcting products and SOA yields to

diminish the wall loss effect.

**3 Results and discussion**

**3.1 Wall loss correction**

The wall loss fractions of four kinds of LMW peroxides observed in limonene ozonolysis were discussed here,

i.e., $H_2O_2$, hydroxymethyl hydroperoxide (HMHP), peroxyformic acid (PFA), and peroxyacetic acid (PAA).

Figure 1 showed the dependence of these peroxides wall loss fractions on RH. The four loss profiles indicated

increasing loss fractions of LMW peroxides with increasing RH, and this tendency was especially obvious for

PFA and PAA, whose loss fractions increased successively with RH. For $H_2O_2$ and HMHP, their wall loss

fractions went up quickly above 50% RH, yet didn't have large change below 50% RH. Generally speaking,

HMHP had the highest wall loss fraction, which could reach ~ 0.25 at 90% RH, while PFA had the lowest wall

loss fraction. The average of these LMW peroxides loss fractions was used to correct the wall loss effect for

gas-phase HMW peroxides.

The wall loss effect on a series of carbonyls formed in limonene ozonolysis was evaluated in the RH range of 0–

90%, and the profiles as a function of RH were shown in Fig. 2. In general, the relationship of carbonyls loss

fractions with RH was not very obvious. The loss curves of formaldehyde (FA), acetaldehyde (AA), and acetone

(ACE) were kind of irregular. For hydroxyacetone (HACE), glyoxal (GL), and methylglyoxal (MGL), their wall



loss fractions were lowest at 10% RH, and then they gradually arose with increasing water vapor concentration. The higher loss fractions of GL and MGL compared with other carbonyls could be attributed to their inclination towards hydration. FA had the lowest wall loss fraction around 0.03, while MGL had the highest loss fraction around 0.12.

The wall loss effect on SOA was also discussed from 0% to 90% RH. The SOA wall loss fraction was ~ 0.06 at dry condition, then increased slightly with water vapor concentration. At 50% RH the SOA loss fraction was ~ 0.11, and at 90% RH, the SOA loss fraction was ~ 0.17. Detailed information was shown in the Supplement.

**3.2 SCIs generation**

The reaction channels of ECIs are complex in monoterpene ozonolysis, and some studies suggested that the
unimolecular decomposition and intermolecular stabilization were dominant pathways (Aschmann et al., 2002; Chew and Atkinson, 1996; Lin et al., 2014; Ma et al., 2008; Tillmann et al., 2010). Due to the relatively long lifetime of SCIs, bimolecular reactions of SCIs with other trace species are possible. The reaction between SCIs and water has received much attention because sufficient water vapor exists in the atmosphere, thus it is regarded as an important reaction pathway for SCIs. This class of reaction, which produces α-hydroxyalkyl
hydroperoxides (HAHPs) decomposing to $H_2O_2$, carbonyls, and carboxylic acids, is thought to be an essential source of these compounds and serve as a principle source of $H_2O_2$ formation without light (Becker et al., 1990; Becker et al., 1993; Gäb et al., 1985; Sauer et al., 1999). However, because of the complicated structure and the difficulty of synthesis, it is not easy to observe monoterpene SCIs reaction directly, resulting in that the reaction mechanism and the rate constant of monoterpene SCIs reaction with water are still unclear. Vereecken et al.
(2017) reported that the concentration of biogenic SCIs in the atmosphere was strongly limited by unimolecular decay, yet the obvious formation of products from HAHPs decomposition under humid condition in



monoterpene ozonolysis demonstrated that SCIs reaction with water was important (Anglada et al., 2002; Ma et al., 2008; Tillmann et al., 2010). Jiang et al. (2013) suggested that the formation of HAHPs was the most favorable pathway for limonene SCIs reaction with $H_2O$, and the subsequent decomposition of HAHPs was

thought to be prior to generate aldehyde and $H_2O_2$ (Chen, 2016; Kumar, 2014). Although theoretical calculation results indicated that HAHPs decomposition was slow, some studies proved that water and acid molecule could greatly promote the decomposition process (Anglada et al., 2002; Anglada et al., 2011; Aplincourt and Anglada, 2003; Crehuet et al., 2001), and the $H_2O_2$ formation from HAHPs decomposition was very fast (Chen et al., 2016; Winterhalter et al., 2000). The fact that few HAHPs larger than HMHP were identified in alkene

ozonolysis also provided evidence for the rapid decomposition of large HAHPs. In this study, through investigating hydroperoxides formation from 0% to 90% RH in limonene ozonolysis, we tried to provide more information about limonene SCIs generation and their reaction with water.

Figure 3 showed the dependence of $H_2O_2$ and HMHP yield on RH, and the six profiles in each subgraph represented conditions at low or high $[O_3]/[limonene]$ in the presence or absence of OH scavenger (2-butanol or

cyclohexane). The molar yield used here was defined as the ratio of products molar number to the molar number of limonene consumed. It was obvious that although in different cases, the variation of $H_2O_2$ yield and HMHP yield with RH had similar tendency, both of which increased significantly from 0% to 70% RH, and then they approached the limiting values. The effect of OH scavenger was not obvious. At low $[O_3]/[limonene]$, the maximum $H_2O_2$ yield was ~ 24.00% without OH scavenger, ~ 24.60% with 2-butanol, and ~ 22.95% with

cyclohexane, respectively. At high $[O_3]/[limonene]$, the maximum $H_2O_2$ yield reached ~ 41.20% without OH scavenger, ~ 41.80% with 2-butanol, and ~ 40.50% with cyclohexane. As for HMHP, its yield was much higher at high $[O_3]/[limonene]$ (~ 5.43%) than at low $[O_3]/[limonene]$ (~ 0.62%), and the specific information could be found in the following picture. It is usually believed that $H_2O_2$ has two generation pathways, one is $HO_2$





self-reaction, and the other one is HAHPs decomposition. The former pathway is considered as a main

contributor to $H_2O_2$ during daytime, while the latter is regarded as a route without photochemistry. In the

reaction system discussed here, when we applied a box model coupled with limonene reaction mechanism

extracted from the Master Chemical Mechanism (MCM) v3.3 (website: http://mcm.leeds.ac.uk/MCMv3.3.1) to

simulate the reaction, it was estimated that the yield of $H_2O_2$ formed from $HO_2$ self-reaction was less than 0.1%

under both dry and wet conditions. When 2-butanol or cyclohexane reaction mechanism was taken into

consideration the contribution of this pathway to $H_2O_2$ formation was still very limited, hence, it was assumed

that $HO_2$ self-reaction was not important for $H_2O_2$ generation in this reaction system. Under dry condition, a

small amount of water vapor might desorb from the flow tube wall and participated in reactions resulting in a

little $H_2O_2$ formation. From 0% to 70% RH, both of $H_2O_2$ yield and HMHP yield were promoted significantly

by the increasing RH, indicating that the reaction with water gradually turned to be dominant for limonene SCIs.

Above 70% RH, the appearance of the limiting values of $H_2O_2$ yield and HMHP yield suggested that the water

vapor concentration was high enough to make the bimolecular reaction of SCIs and $H_2O$ suppress other reaction

channels. The results here proved that reaction with water was an essential route for limonene SCIs, and the

rapid decomposition of HAHPs made an important contribution to $H_2O_2$ formation.

According to experimental conditions elaborated in Sect. 2.2, we tried to calculate the contribution of

endocyclic DB and exocyclic DB ozonolysis to $H_2O_2$ and HMHP formation, and furthermore, infer the SCIs

generation in different DBs ozonolysis. The predicted SCIs yield was derived by combining the limiting yields

of $H_2O_2$ and HMHP together, based on the assumption that the limiting yield of $H_2O_2$ was equal with the large

SCIs yield (Hasson, 2001a, b). The SCIs yield estimated here could be regarded as a lower bound as a small

fraction of SCIs might undergo decomposition. It was observed that OH scavenger didn't have a huge impact on

the SCIs measurement results, while big difference existed between the two DBs ozonolysis. The SCIs yield of



endocyclic DB ozonolysis was around 24.45%, yet the exocyclic DB ozonolysis had larger stabilization fraction

of ECIs, which was about 42.90%. It meant that even though exocyclic DB ozonolysis was much slower than

endocyclic DB, it played an unneglibile role in generating SCIs in limonene ozonolysis.

### 3.3 OH radical generation

In the last few years, OH formation pathways in alkene ozonolysis were extensively studied, and the major

pathway was considered to be the unimolecular decomposition of ECIs. Besides, some studies suggested that

SCIs could also generate OH radical through self-decomposition or reaction with other species (Anglada et al.,

2002; Hasson et al., 2003; Kroll et al. 2001a, b; Tillmann et al., 2010; Zhang and Zhang, 2005). When it came to

OH formation in alkene ozonolysis, one thing still under debate was the effect of water because of the large

discrepancy among the existing publications. Anglada et al. (2002) indicated that water could increase OH

production using quantum mechanical calculations. Nevertheless, observed influence of RH on OH production

was limited. Tillmann et al. (2010) reported higher OH yield under humid condition, but more studies showed

that OH formation was independent of water vapor concentration (Aschmann et al., 2002; Atkinson and

Aschmann, 1993; Atkinson et al., 1992; Berndt et al., 2003; Forester and Wells, 2011; Hasson et al., 2003). OH

yield in alkene ozonolysis is highly dependent on the reactant molecular structure, and as far as we know, only

Herrmann et al. (2010) researched OH yield of both DBs in limonene ozonolysis under dry condition. OH

radical could be directly detected by laser-induced fluorescence (LIF), or be indirectly determined using OH

radical scavenger. 2-butanol and cyclohexane are both commonly-used OH scavengers, and OH yield can be

determined by detecting the amount of 2-butanone generated from 2-butanol reaction with OH or the amount of

cyclohexanone plus cyclohexanol formed from cyclohexane reaction with OH. Aschmann et al. (2002)

suggested that using 2-butanol to measure OH formation would be more accurate than using cyclohexane.



In this study, OH yield was determined by adding sufficient 2-butanol as scavenger and measuring how much 2-butanone generated. The yield of 2-butanone formed from the reaction of 2-butanol and OH had been detected previously (Aschmann et al., 2002; Baxley and Wells, 1998; Chew and Atkinson, 1996), and we used 0.66 as an

average of the reported values to calculate OH radical yield as the same as Forester and Wells (2011). The results showed that the yield of OH radical produced from endocyclic DB ozonolysis was $0.65 \pm 0.21$, while for exocyclic DB, OH yield was $0.24 \pm 0.13$. OH generation didn't show evident dependence on water vapor concentration, and the OH yield of endocyclic DB ozonolysis was in the range of values published, while the OH yield of exocyclic DB ozonolysis was determined to be higher than that Herrmann et al. (2010) reported.

The fact that OH yields of both unsaturated bonds were not obviously affected by RH suggested that the major pathway of OH generation in limonene ozonolysis was decomposition of ECIs through hydroperoxide channel. The possibility that other OH formation pathways also existed could not be totally excluded, but we speculated that the contribution of other pathways to OH formation was not significant. Hence it could be concluded that in limonene ozonolysis, the endocyclic DB was inclined to generate OH radical through ECIs decomposition,

while the exocyclic DB had higher fraction of stabilization forming SCIs.

### 3.4 Peroxycarboxylic acids generation

As a species accounting for 40–50% of the total global organic peroxides (Crounse et al., 2006; Khan et al., 2015), peroxycarboxylic acids (RC(O)OOH) play an important role in promoting atmospheric oxidation capacity and enhancing the acidity of aqueous phase. Here the generation of PFA (CH(O)OOH) and PAA

($CH_3C(O)OOH$) were observed in limonene ozonolysis. These two kinds of peroxycarboxylic acids got growing attention recent years as reactive oxidants, and PFA had shown to be more active than PAA. In field observations, PAA was reported widely in remote and urban areas (Lee et al., 2000; Liang et al., 2013; Zhang et

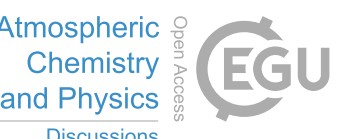

al., 2010), yet there were few reports about PFA existence in the atmosphere (Liang et al., 2015), which might

be attributed to the rapid decomposition of PFA or its precursor. The dominant formation pathway of PAA was

considered as the reaction of $CH_3C(O)OO$ radical with $HO_2$ (Groβ et al., 2014; Lightfoot et al., 1992; Winiberg

et al., 2016), while the formation pathway of PFA hadn't been identified yet and was speculated to be the

reaction of $HC(O)OO$ radical with $HO_2$ (Liang et al., 2015). In this study the yields of PFA and PAA produced

in limonene ozonolysis both showed large discrepancy between the low and high ratio sets of experiments (Fig.

4). When the exocyclic DB was ozonated, the generation of PFA and PAA were enhanced to a large extent

compared with only endocyclic DB ozonolysis, indicating that exocyclic DB ozonolysis had an important

impact on PFA and PAA. As far as we know, this was the first time that such high yields of PFA and PAA in

alkene ozonolysis were reported. For both of PFA and PAA, the highest molar yields were observed when no

OH scavenger was used, demonstrating that OH reaction contributed to part of their formation. When OH

radical was scavenged, we speculated that PFA and PAA formation were mainly through ECIs isomerization and

decomposition following $HC(O)OO$ and $CH_3C(O)OO$ radicals generation. These radicals could further react

with $HO_2$ forming peroxycarboxylic acids, which provided a plausible explanation for higher yields of PFA and

PAA observed in experiments with 2-butanol than with cyclohexane.

The yields of PFA and PAA had positive relationship with water vapor concentration in all the cases, reaching

the highest level at 90% RH. Model simulation results significantly underestimated the PAA formation and

ignored the RH effect, indicating some missing pathways of PAA generation related with water. Because of the

deficiency of PFA mechanism in MCM we didn't simulate the PFA formation, however, the positive correlation

between the yields of PFA and PAA could provide some evidence for the assumption that they might have

similar forming mechanism. The rate constant of $RO_2$ reaction with $HO_2$ had been investigated in a series of

studies and was thought to be unaffected by water (Atkinson et al., 1999; Lightfoot et al., 1992; Tyndall et al.,

2001; Wallington et al., 1992). Therefore the increase of PFA and PAA yields with RH might be attributed to the promoting effect of water on $HC(O)OO$ and $CH_3C(O)OO$ radicals generation. Because the highly reactive peroxides were easy to lose on the wall, studies in chamber, which usually last for hours, might be hard to observe these peroxycarboxylic acids. The results here demonstrated that limonene ozonolysis could contribute to PFA and PAA formation, and although the high instability and reactivity made PFA difficult to observe, it

might have a short stay in the atmosphere.

**3.5 Particulate peroxides and $H_2O_2$ generation**

**3.5.1 $H_2O_2$ evolution at low $[O_3]/[limonene]$**

$H_2O_2$ generation from SOA in aqueous phase was thought to be a possible way of producing $H_2O_2$ in cloud water or releasing $H_2O_2$ continuously after inhalation. Here we provided the quantitative measurement of $H_2O_2$

generation from SOA produced in limonene ozonolysis in different cases. Considering that when SOA extract solution was kept at room temperature (298 K) some unstable constituents would decay rapidly resulting in that the evolution process couldn't be observed completely, we chose to keep the SOA solution at 277 K for several days in the dark to determine the change of total peroxides concentration and $H_2O_2$ concentration successively. In the low ratio set of experiments, the concentration of total peroxides in solution nearly maintained stable in

48 h and $H_2O_2$ concentration could reach a steady state after going through a short rising period. After aqueous $H_2O_2$ concentration reached plateau, the amount of particulate $H_2O_2$ per particle mass formed in different cases from 0% to 90% RH was calculated and shown in Table 2. No matter OH scavenger was used or not, SOA produced at higher RH was inclined to own higher capacity of producing $H_2O_2$. When no OH scavenger was used, the lowest $H_2O_2$ content per particle mass was ~ 1.13 ng/μg at dry condition, and the highest value was ~

2.45 ng/μg at 90% RH. When 2-butanol was used the changing trend of $H_2O_2$ generation resembled the

condition without scavenger, and the minimum was ~ 1.33 ng/μg at 0% RH, while the maximum was ~ 2.89

ng/μg at 80% RH, which was a little higher than that at 90% RH. It was interesting to note that, in the presence

of cyclohexane, the trend of $H_2O_2$ generation in SOA solution differed greatly from both of the above. Even

under dry condition, the $H_2O_2$ content per particle mass could reach ~ 3.22 ng/μg, and the maximum was ~ 4.63

ng/μg at 90% RH.

### 3.5.2 $H_2O_2$ evolution at high [$O_3$]/[limonene]

In the experiments of high [$O_3$]/[limonene], SOA produced with different scavengers was found to have

different rates of generating $H_2O_2$ in solution at 277 K, according to which we determined appropriate detection

frequency and total duration for the three kinds of SOA. SOA produced without OH scavenger had the lowest

rate of generating $H_2O_2$, so an eight-day measurement result was reported here. For SOA produced in the

presence of 2-butanol, it was a little faster than the former on generating $H_2O_2$ and reached the limiting value

within six days. However, SOA produced with cyclohexane had a much faster rate than both of the above, and

the $H_2O_2$ concentration in solution became stable within three days. For all of the three kinds of SOA, the total

peroxides concentration decreased slightly in the analysis duration, which was concretely clarified in the

Supplement.

It was obvious that SOA produced in the high ratio set of experiments had greater ability of generating $H_2O_2$,

and Fig. 5 showed the time profiles of $H_2O_2$ evolution of different kinds of SOA. When no OH scavenger was

used, $H_2O_2$ concentration in solution rose constantly in about six days then became stable. The limiting value of

$H_2O_2$ generation was influenced by RH, which was ~ 5.28 ng/μg at 0% RH, then increased gradually with

increasing RH until 80% RH (~ 14.45 ng/μg). In the experiments with 2-butanol, $H_2O_2$ concentration kept rising

in the first four days then became stable. At dry condition, the limiting value of $H_2O_2$ content was ~ 7.60 ng/μg,





then it increased until 50% RH. SOA produced above 50% RH didn't show obvious difference, and the highest

$H_2O_2$ content was ~ 15.50 ng/µg. When cyclohexane was added, $H_2O_2$ concentration in solution rose up quickly

in the first day then tended to be stable. The limiting value of $H_2O_2$ generation was also affected by water vapor

concentration, yet the promoting effect was not very significant and no big difference was observed above 30%

RH. The limiting $H_2O_2$ content per particle mass was ~ 16.64 ng/µg at dry condition and was ~ 30.00 ng/µg

above 30% RH.

**3.5.3 Different stabilities of particulate peroxides**

According to the measurement results of total peroxides and $H_2O_2$ in SOA solution, the particulate peroxides

could be roughly divided into two categories: one was unstable that could decompose or hydrate to generate

$H_2O_2$, the other one was stable that could nearly remain unchanged in several days. Based on the assumption

that all the peroxides contained one peroxy bond, the molar fractions of peroxides with different stabilities under

various conditions could be calculated. The molar fraction used here represented the molar number of stable or

unstable particulate peroxides to the molar number of total peroxides in SOA. At low [O$_3$]/[limonene], the molar

fraction of unstable particulate peroxides was around 0.11–0.13 in the case of adding no scavenger, which was

similar with the case of adding 2-butanol. In the presence of cyclohexane, the molar fraction of unstable

particulate peroxides would increase to 0.20–0.32. At high [O$_3$]/[limonene], the molar fractions of unstable

peroxides in the case of adding no scavenger and adding 2-butanol were also similar, both of them ranged in

0.13–0.25, yet this value would reach ~ 0.50 when cyclohexane was used. Detailed information was shown in

the Supplement. Model results showed that most peroxides produced in the reaction were ROOH and R(O)OOH,

in addition, some studies proposed that peroxyhemiacetals also made important contribution to aerosol (Tobias

and Ziemann, 2000; Tobias et al., 2000). The acetal reaction producing peroxyhemiacetals was reversible, so



part of the peroxyhemiacetals in SOA were possible to hydrolyze and form some peroxides. To investigate the

stabilities of ROOH and R(O)OOH, we synthesized methyl hydroperoxide (MHP) and ethyl hydroperoxide

(EHP) to represent ROOH. As for R(O)OOH, PFA and PAA were used as representative. All the synthesis

solutions were stored at 277 K, which was the same with the experimental condition. PFA and PAA were found

to decompose and generate $H_2O_2$ in several days, yet MHP and EHP maintained stable. Hence we speculated

that peroxycarboxylic acids and peroxyhemiacetals might be the main components of unstable particulate

peroxides and contribute to $H_2O_2$ generation in aqueous phase.

SOA formed with different OH scavengers had different molar fractions of unstable peroxides, which indirectly

proved the influence of radical chemistry on SOA composition. When cyclohexane was used as OH scavenger,

the formation of more unstable species might be attributed to the extra $RO_2$ radicals provided by cyclohexane

that participated in subsequent reactions. The amount of $H_2O_2$ generation at low $[O_3]/[limonene]$ measured here

was comparable with the published value of $H_2O_2$ produced from α-pinene SOA in solution (Li et al., 2016;

Wang et al., 2011). However, at high $[O_3]/[limonene]$, $H_2O_2$ generation level increased significantly, which

proved that the multigeneration oxidation improved the formation of peroxycarboxylic acids and

peroxyhemiacetals in particles. The results demonstrated that SOA produced in limonene ozonolysis could

behave as an important source of $H_2O_2$ in aqueous phase, and they also showed the difference between the SOA

formed from single-unsaturated monoterpene ozonolysis and double-unsaturated monoterpene ozonolysis.

**3.6 Contribution of peroxides to SOA**

**3.6.1 SOA formation**

The yield of SOA produced from limonene ozonolysis in different cases from 0% to 90% RH was measured.

The SOA yield was defined as the ratio of aerosol mass concentration to the mass concentration of limonene



consumed. In the six different cases, SOA yield was determined to be unaffected by water, yet it showed strong
dependence on the reactants ratio and the use of OH scavenger, which was shown in Table 3. Whether at low or
high $[O_3]/[limonene]$, the case of none OH scavenger had the highest SOA yield, which suggested the effect of
OH reaction on aerosol formation, and the order of SOA yield was no scavenger > 2-butanol > cyclohexane. The
case owning the highest SOA yield was high $[O_3]/[limonene]$ without OH scavenger ($0.511 \pm 0.097$), while the
lowest SOA yield was detected at low $[O_3]/[limonene]$ with cyclohexane ($0.288 \pm 0.038$).

**3.6.2 Peroxides mass fraction**

In this study, the iodometric method was used to analyze the particle-phase total peroxides content in limonene
ozonolysis. The iodometric method was commonly regarded as a standard method of detecting total peroxides
and it could almost quantify all kinds of peroxides (Bonn et al., 2004; Jenkin, 2004). The mass fraction used
here was defined as the ratio of particulate peroxides mass to SOA mass. The average molecular weight of

peroxides in particles was assumed to be 300 g/mol, and the mass fractions of peroxides in the six cases from 0%
to 90% RH were summarized in Table 3. In each case, the peroxides mass fraction in SOA increased slightly
with RH, and significant difference existed between the low and high ratio sets of experiments. When no OH
scavenger was used, the range of peroxides mass fraction was 0.065–0.169 at low $[O_3]/[limonene]$, and 0.401–
0.492 at high $[O_3]/[limonene]$. When using different scavengers, some changes were observed. In the presence

of 2-butanol, peroxides mass fraction ranged in 0.101–0.189 at low $[O_3]/[limonene]$, and in 0.502–0.580 at high
$[O_3]/[limonene]$. In the presence of cyclohexane, peroxides mass fraction was in the range of 0.087–0.189 at low
$[O_3]/[limonene]$, and 0.477–0.512 at high $[O_3]/[limonene]$. Docherty et al. (2005) reported that peroxides mass
fraction in α-pinene SOA was ~ 0.47, while for β-pinene SOA the fraction was ~ 0.85. Li et al. (2016) observed
that peroxides could account for ~ 0.21 in α-pinene SOA. Here we first reported the mass fraction of peroxides

in SOA derived from limonene ozonolysis, highlighting the important role of organic peroxides in SOA

composition, especially when multigeneration oxidation happened.

**3.6.3 Peroxides partitioning**

The molar yields of HMW peroxides in both gas-phase and particle-phase were determined in different cases.

The effect of RH and OH scavenger was not obvious, while under high [O$_3$]/[limonene] the HMW peroxides

yield increased by ~ 10% in contrast with the condition of low [O$_3$]/[limonene]. Furthermore, the gas-phase

HMW peroxides and particle-phase HMW peroxides were discussed separately, which was shown in Fig. 6. For

gas-phase HMW peroxides, the molar yield showed a decreasing tendency with increasing RH, which was

obvious at high [O$_3$]/[limonene]. For particle-phase HMW peroxides, the molar yield showed a slight increasing

dependence on RH and their yield was promoted significantly by oxidizing degree. A possible explanation for

the RH effect was that the water content in aerosol would increase when RH increased, and promote the uptake

of some compounds into particles. However, some researches proved that water effect on the partitioning of

organic products seemed to be small especially when no inorganic seed particles were used (Jonsson et al., 2006,

2008a, b), so we proposed that the promoting effect of water on particulate HMW peroxides could be mainly

attributed to the chemical effect. The contribution of multigeneration-oxidation products to SOA formation in

limonene ozonolysis had been stated by some studies (Hoffmann et al., 1997; Ng et al., 2006), and the results

here indicated that organic peroxides might account for a considerable proportion of those products. On the one

hand, multigeneration oxidation helped produce more low-volatility peroxides in the gaseous phase that could

partition into particles, and on the other hand, it could also accelerate the occurrence of some heterogeneous

reactions through providing more reactive species.




### 3.7 Contribution of carbonyls to SOA

### 3.7.1 Carbonyls formation

Carbonyl compounds including HACE, FA, AA, ACE, GL, and MGL were detected in the reaction. Unlike peroxides, whose generation often showed obvious increasing dependence on RH, only FA yield would increase with RH and other carbonyls production were not significantly affected by water. The yields of HACE, AA, ACE, GL, and MGL in different cases were summarized in Table 4. The fact that HACE formed without OH scavenger but did not form when cyclohexane was used indicated that HACE might generate from OH reactions. As for AA and ACE, both of their yields in the presence of cyclohexane were found to be lower than the case without scavenger, especially at low [O$_3$]/[limonene]. This suggested that OH reaction contributed to a potion of their formation and the endocyclic DB ozonolysis did not tend to generate AA and ACE. It was speculated that these three kinds of carbonyls could also generate from the reaction of 2-butanol or HO$_2$ radicals promoted their formation, so the presence of 2-butanol increased their yields. As regards GL and MGL, RH and OH scavenger did not make big influence, and both of their yields at high [O$_3$]/[limonene] were higher than the condition of low [O$_3$]/[limonene]. The generation of FA had a positive dependence on RH, which was specifically illustrated in the Supplement and the total variation range could be found in Table 4.

### 3.7.2 Experimental and theoretical partitioning coefficients

A parameter that has been used widely to describe the partitioning feature of a compound is described as follows (Pankow, 1994; Pankow and Bidleman, 1992):

$$K_{p,i} = \frac{F_i/TSP}{A_i} \qquad (1)$$

Where $K_{p,i}$ (m$^3$ μg$^{-1}$) is the partitioning coefficient of compound i, $TSP$ (μg m$^{-3}$) is the concentration of total



suspended particulate matter, $F_i$ (μg m⁻³) and $A_i$ (μg m⁻³) are the particulate and gaseous concentrations of

compound i, respectively. The measured partitioning coefficients of FA, AA and ACE were on the magnitude of

$10^{-5}$, and the partitioning coefficients of HACE, GL, and MGL were on the magnitude of $10^{-4}$.

Theoretical gas-particle partitioning coefficients of these compounds were calculated using the absorption

equilibrium equation defined by Pankow (1994), which was used widely to estimate the ability of a substance to

partition into the particulate phase and predict SOA yield in model (Griffin et al., 1999; Hohaus et al., 2015;

Odum et al., 1996; Yu et al., 1999):

$$K_{p,i} = \frac{760\ R\ T\ f_{om}}{MW_{om}\ 10^6\ \zeta_i\ p_{L,i}^0} \tag{2}$$

Where R is the ideal gas constant ($8.206 \times 10^{-5}$ m³ atm mol⁻¹ K⁻¹), T is temperature (K), $f_{om}$ is the mass

fraction of TSP that is the absorbing organic material (om), which is 1 here, $MW_{om}$ (g mol⁻¹) is the mean

molecular weight of om phase, which is estimated to be 130 g mol⁻¹ in this study, $\zeta_i$ is the activity coefficient

of compound i in the om phase, which is assumed to be unity, and $p_{L,i}^0$ (Torr) is the vapor pressure of

compound i, which is predicted by the method of Moller et al. (2008). The calculated coefficients of HACE, FA,

AA, ACE, GL, and MGL were $3.972\times10^{-8}$ m³ μg⁻¹, $2.096\times10^{-11}$ m³ μg⁻¹, $1.624\times10^{-10}$ m³ μg⁻¹, $6.192\times10^{10}$ m³ μg⁻

¹, $5.476\times10^{-10}$ m³ μg⁻¹, and $1.246\times10^{-9}$ m³ μg⁻¹, respectively, at 298K. The experimental $K_p$ value of HACE

was about 10,000 times bigger than the theoretical value, and for other carbonyls, the experimental $K_p$ was

about 100,000 times bigger than the theoretical value. The gap between the experimental $K_p$ and predicted $K_p$

of GL and MGL we estimated was comparable with the results of Healy et al. (2008, 2009), but higher than that

of Ortiz et al. (2013). The fact that gas-particle partitioning coefficients of carbonyls observed were much higher

than theoretical values indicated that carbonyl compounds made a more important contribution for SOA

formation than estimated. Although the partitioning coefficients measured in experiments showed huge





difference with calculated values, some relationship between the measured $K_p$ and the vapor pressure of carbonyl compounds was observed. Figure 7 showed the dependence of measured $lg(K_p)$ and predicted $lg(K_p)$ on $lg(p^0)$, and their linear fitting curves. The slope of the linear fitting equation of predicted $lg(K_p)$ versus $lg(p^0)$ was -0.964, and $R^2$ was 0.998. The slope of the linear fitting equation of measured $lg(K_p)$

versus $lg(p^0)$ was -0.484, and $R^2$ was 0.750, which indicated that the $lg(K_p)$ of carbonyls observed in laboratory also had negative correlation with $lg(p^0)$. A plausible explanation for the large difference between the measured and predicted $K_p$ was that carbonyl compounds were easy to polymerize and react with other species on particles, and the relationship of measured $K_p$ and $p^0$ might provide some reference for predicting the contribution of carbonyls to SOA formation.

**4 Conclusions and implications**

An experimental study about the oxidants transition and SOA property in limonene ozonolysis with respect to the role of different DBs, radical chemistry, and water was reported in this work. To investigate the oxidants transition in this reaction system, a series of products owning the oxidizing capacity including SCIs, OH radical, and peroxides in both gaseous and particulate phases were detected. Based on the variation of $H_2O_2$ and HMHP

generation on RH, the importance of limonene SCIs reaction with water was confirmed and the yield of SCIs could be estimated, which was ~ 0.24 for endocyclic DB and ~ 0.43 for exocyclic DB. OH radical yields of endocyclic and exocyclic DBs were indirectly determined to be ~ 0.65 and ~ 0.24, demonstrating the different reaction mechanisms of different DBs in limonene ozonolysis. The formation of two peroxycarboxylic acids PFA and PAA was observed and their high yields were first reported in alkene ozonolysis. The yields of PFA and

PAA increased with RH and oxidizing degree, showing the effect of water and the exocyclic DB oxidation on their formation. The $H_2O_2$ generation from SOA in solution provided evidence for the ability of SOA to

contribute oxidants in aqueous phase. Particles produced at high $[O_3]/[\text{limonene}]$ showed much higher potential of forming $H_2O_2$ than particles produced at low $[O_3]/[\text{limonene}]$, and the difference in $H_2O_2$ generating rate and amount among particles formed with different OH scavengers demonstrated the impact of radical chemistry on

SOA composition. The partitioning behaviors of peroxides and carbonyls were discussed and the results showed their importance to SOA formation. Particulate peroxides could account for 0.07–0.19 in limonene SOA at low $[O_3]/[\text{limonene}]$ and 0.40–0.58 at high $[O_3]/[\text{limonene}]$, which proved the important role of peroxides in SOA composition especially when multigeneration oxidation happened. The partitioning coefficients of carbonyls observed in laboratory were always several orders of magnitude higher than theoretical values, since their ability

to polymerize and react with other species on particles their contribution to SOA was higher than the estimation. Through determining the formation of oxidizing species and the partitioning of peroxides and carbonyls, the issues of oxidants transition and SOA property in limonene ozonolysis were investigated. Limonene showed its specificity in many aspects because of its different DBs, suggesting that some influence of terpenes containing more than one DB in the atmosphere might be underestimated before. The results demonstrated the effect of

oxidizing degree, radical chemistry, and water on limonene SOA property, while the structures and properties of particulate products especially peroxides should be studied further. The implications of limonene chemistry in the atmosphere were close with the multigeneration oxidation that happened on its different DBs, and whether the phenomena could be observed in other terpenes oxidation that contained more than one DB needed more laboratory evidence.

**Acknowledgements**

We gratefully acknowledge the National Natural Science Foundation of China (grants 21477002) and the National Key Research and Development Program of China (grants 2016YFC0202704) for financial support.



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



**Table 1.** Experimental Conditions.

| Exp. | [Limonene] (ppbv) | [O$_3$] (ppbv) | OH Scavenger | [OH Scavenger] (ppmv) | RH (%) |
|---|---|---|---|---|---|
| L(No-sca) | 280 | 500 | — | — | 0–90 |
| L(2-But) | 280 | 500 | 2-butanol | 350 | 0–90 |
| L(C-hex) | 280 | 500 | cyclohexane | 420 | 0–90 |
| H(No-sca) | 183 | 19000 | — | — | 0–90 |
| H(2-But) | 183 | 19000 | 2-butanol | 350 | 0–90 |
| H(C-hex) | 183 | 19000 | cyclohexane | 420 | 0–90 |

Note: L, low ratio; H, high ratio; No-sca, none scavenger; 2-But, 2-butanol; C-hex, cyclohexane; RH, relative humidity.



**Table 2.** $H_2O_2$ generation per particle mass (ng/ug) in SOA formed with different OH scavengers in the relative humidity

(RH) range of 0–90% under low $[O_3]/[limonene]$ ratio.

|        | 0% RH      | 10% RH     | 30% RH     | 50% RH     | 70% RH     | 80% RH     | 90% RH     |
|--------|------------|------------|------------|------------|------------|------------|------------|
| No-sca | 1.13±0.22  | 1.42±0.40  | 1.53±0.28  | 1.88±0.16  | 2.24±0.42  | 2.26±0.44  | 2.45±0.48  |
| 2-But  | 1.33±0.15  | 1.56±0.18  | 1.77±0.12  | 2.02±0.65  | 2.55±0.43  | 2.89±0.42  | 2.66±0.57  |
| C-hex  | 3.22±0.52  | 3.95±0.43  | 4.12±0.40  | 4.22±0.33  | 4.63±0.24  | 4.26±0.33  | 4.63±0.96  |

Note: No-sca, none scavenger; 2-But, 2-butanol; C-hex, cyclohexane.



**Table 3.** The SOA yield and mass fraction of particulate peroxides at low or high [O$_3$]/[limonene] ratio in the presence or

absence of OH scavenger from 0% to 90% relative humidity (RH).

|  | L(No-sca) | L(2-But) | L(C-hex) | H(No-sca) | H(2-But) | H(C-hex) |
|---|---|---|---|---|---|---|
| 0% RH | 0.065±0.006 | 0.101±0.009 | 0.087±0.011 | 0.401±0.016 | 0.502±0.008 | 0.477±0.010 |
| 10% RH | 0.091±0.010 | 0.124±0.013 | 0.113±0.009 | 0.436±0.009 | 0.534±0.009 | 0.502±0.013 |
| 30% RH | 0.125±0.010 | 0.147±0.011 | 0.143±0.015 | 0.458±0.020 | 0.553±0.015 | 0.506±0.011 |
| 50% RH | 0.149±0.007 | 0.174±0.011 | 0.161±0.016 | 0.466±0.016 | 0.571±0.009 | 0.512±0.007 |
| 70% RH | 0.155±0.009 | 0.178±0.009 | 0.169±0.014 | 0.486±0.023 | 0.576±0.010 | 0.503±0.011 |
| 80% RH | 0.169±0.013 | 0.189±0.010 | 0.189±0.012 | 0.492±0.015 | 0.580±0.013 | 0.502±0.014 |
| 90% RH | 0.156±0.010 | 0.183±0.013 | 0.189±0.013 | 0.492±0.017 | 0.580±0.018 | 0.506±0.016 |
| SOA Yield | 0.379±0.039 | 0.337±0.048 | 0.288±0.038 | 0.511±0.097 | 0.479±0.044 | 0.401±0.068 |

Note: L, low ratio; H, high ratio; No-sca, none scavenger; 2-But, 2-butanol; C-hex, cyclohexane.



**Table 4.** Yields (%) of carbonyls at low or high [O$_3$]/[limonene] ratio in the presence or absence of OH scavenger.

|  | HACE | FA | AA | ACE | GL | MGL |
|---|---|---|---|---|---|---|
| L(No-sca) | 2.04±0.48 | 7.02±0.90–10.58±0.94 | 1.32±0.24 | 0.22±0.15 | 0.89±0.25 | 0.56±0.34 |
| L(2-But) | 3.94±0.55 | 4.90±0.86–7.77±0.86 | 3.98±0.60 | 0.35±0.18 | 0.69±0.25 | 0.55±0.44 |
| L(C-hex) | — | 5.21±0.66–8.25±0.55 | — | — | 0.81±0.24 | 0.67±0.56 |
| H(No-sca) | 4.45±0.52 | 13.11±0.63–27.00±1.56 | 2.16±0.84 | 0.79±0.22 | 1.33±0.41 | 1.35±0.61 |
| H(2-But) | 10.15±2.11 | 11.03±0.77–23.33±0.62 | 7.86±1.32 | 1.00±0.28 | 1.25±0.36 | 1.31±0.21 |
| H(C-hex) | — | 10.80±1.28–23.32±1.21 | 1.89±1.22 | 0.60±0.47 | 1.25±0.51 | 1.23±0.22 |

Note: —, below detection limit; L, low ratio; H, high ratio; No-sca, none scavenger; 2-But, 2-butanol; C-hex, cyclohexane;

HACE, hydroxyacetone; FA, formaldehyde; AA, acetaldehyde; ACE, acetone; GL, glyoxal; MGL, methylglyoxal.





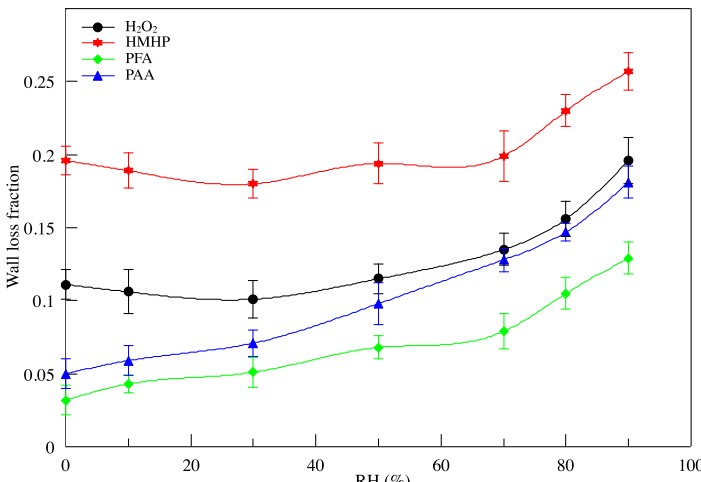

**Figure 1.** The dependence of peroxides wall loss fractions on relative humidity (RH). $H_2O_2$, hydrogen peroxide; HMHP,

hydroxymethyl hydroperoxide; PFA, peroxyformic acid; PAA, peroxyacetic acid.




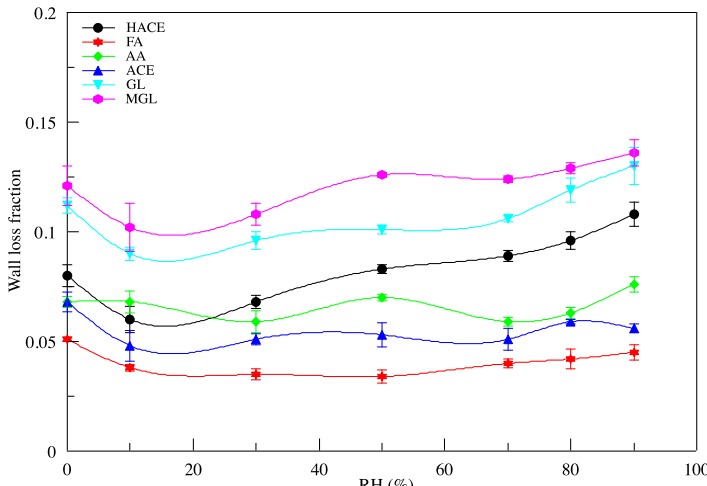

**Figure 2.** The dependence of carbonyls wall loss fractions on relative humidity (RH). HACE, hydroxyacetone; FA, formaldehyde; AA, acetaldehyde; ACE, acetone; GL, glyoxal; MGL, methylglyoxal.

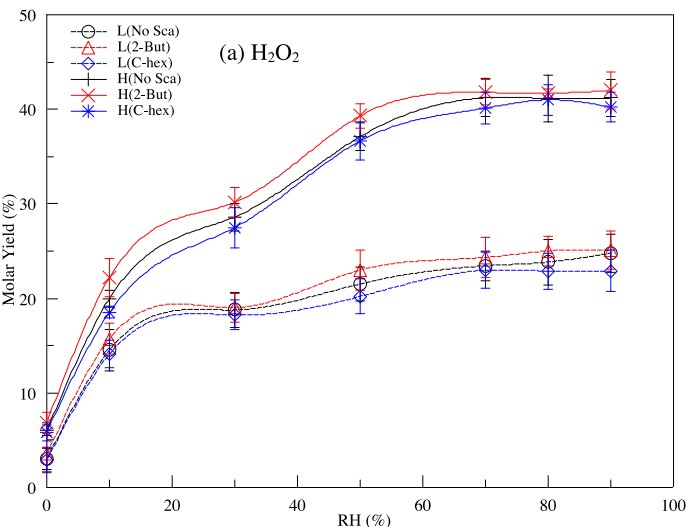


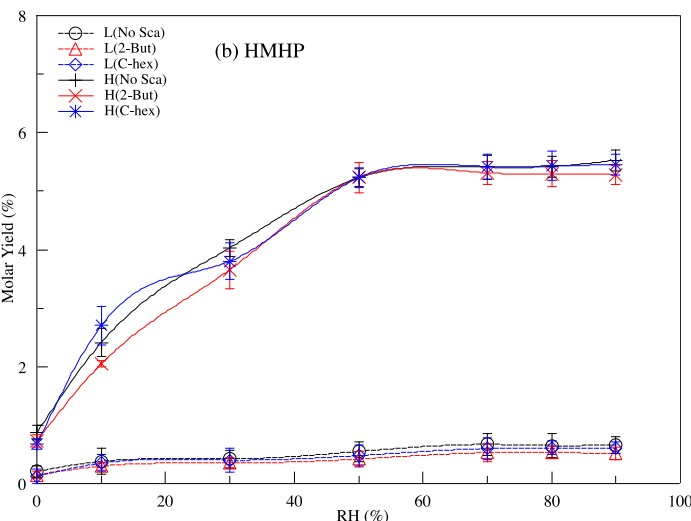

**Figure 3.** Dependence of (a) $H_2O_2$ yield and (b) HMHP yield on relative humidity (RH) at low or high $[O_3]/[limonene]$ ratio

in the presence or absence of OH scavenger (2-butanol or cyclohexane). $H_2O_2$, hydrogen peroxide; HMHP, hydroxymethyl

hydroperoxide; L, low ratio; H, high ratio; No-sca, none scavenger; 2-But, 2-butanol; C-hex, cyclohexane.




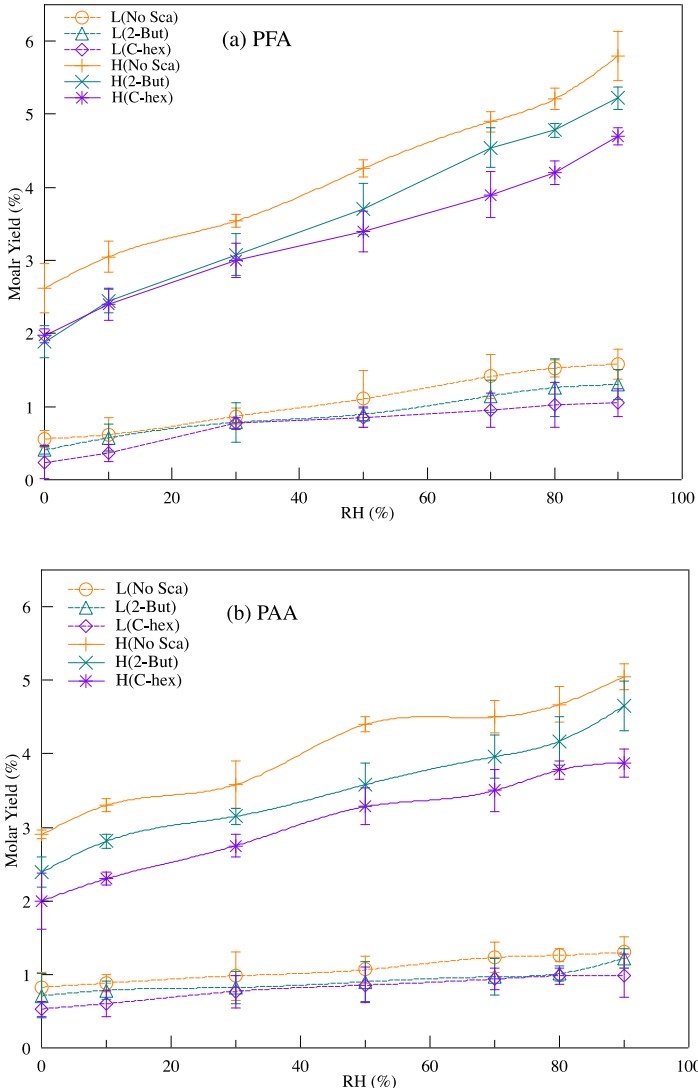

**Figure 4.** Dependence of (a) PFA yield and (b) PAA yield on relative humidity (RH) at low or high [O$_3$]/[limonene] ratio in

the presence or absence of OH scavenger (2-butanol or cyclohexane). PFA, peroxyformic acid; PAA, peroxyacetic acid; L,

low ratio; H, high ratio; No-sca, none scavenger; 2-But, 2-butanol; C-hex, cyclohexane.





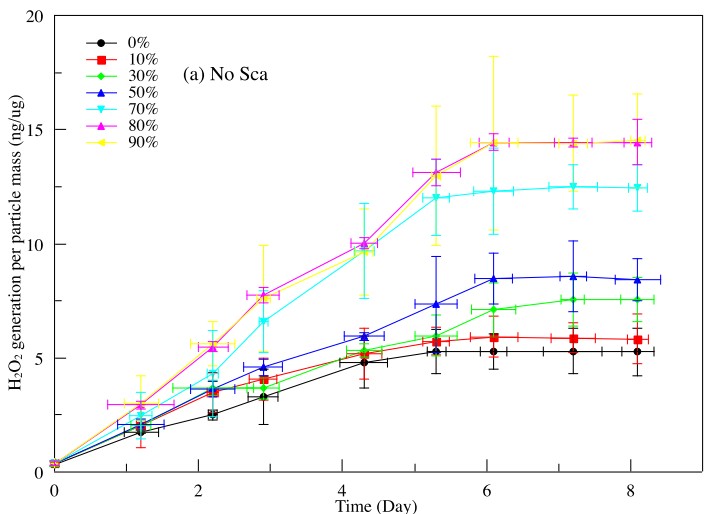

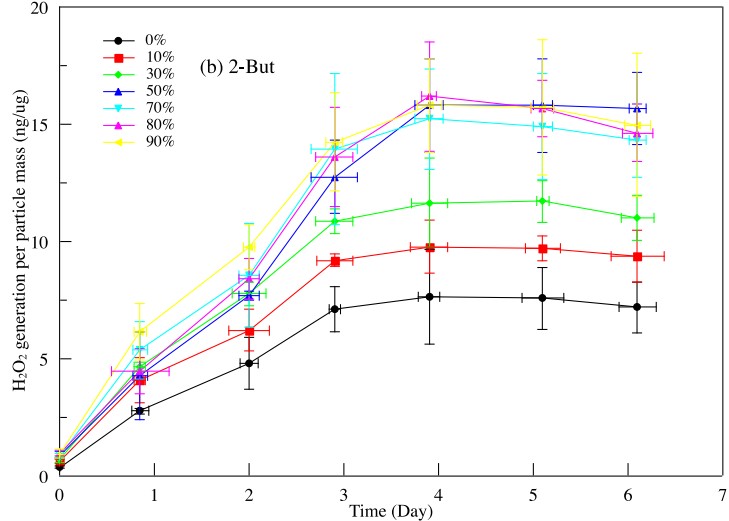





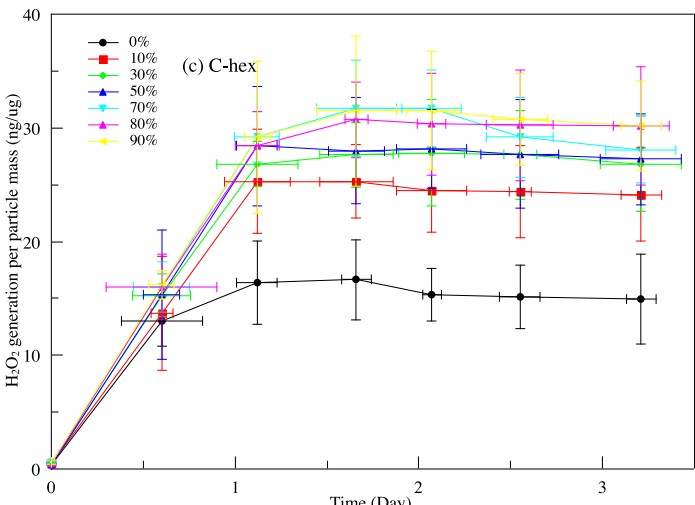

**Figure 5.** Time profiles of $H_2O_2$ evolution per particle mass of different SOA formed (a) without OH scavenger, (b) with 2-butanol, and (c) with cyclohexane in the relative humidity (RH) range of 0–90% under high $[O_3]/[limonene]$ ratio. $H_2O_2$, hydrogen peroxide; No-sca, none scavenger; 2-But, 2-butanol; C-hex, cyclohexane.





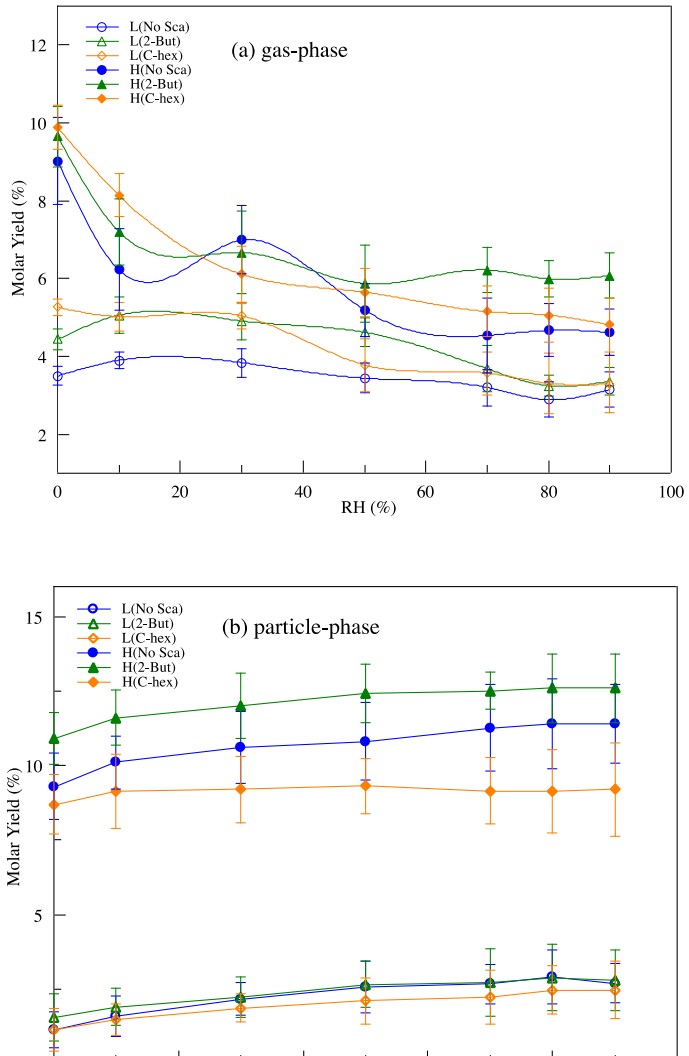


**Figure 6.** The variation of (a) gas-phase and (b) particle-phase high-molecular-weight peroxides molar yields with relative

humidity (RH) at low or high [O$_3$]/[limonene] ratio in the presence or absence of OH scavenger (2-butanol or cyclohexane).

L, low ratio; H, high ratio; No-sca, none scavenger; 2-But, 2-butanol; C-hex, cyclohexane.






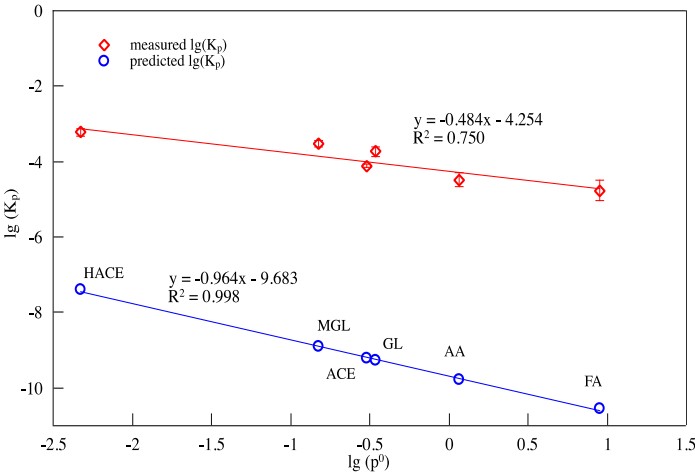

**Figure 7.** The relationship of measured and predicted partitioning coefficients ($K_p$) versus vapor pressure ($p^0$) of carbonyls produced in limonene ozonolysis. HACE, hydroxyacetone; FA, formaldehyde; AA, acetaldehyde; ACE, acetone; GL, glyoxal; MGL, methylglyoxal.