# Peer review of "The oxidation regime and SOA composition in limonene ozonolysis: Roles of different double bonds, radicals, and water"

_Atmospheric Chemistry and Physics, 2018_

## Referee Comment (RC1) · Anonymous Referee #1 · 7 Jun 2018

This paper deals with the production of peroxides and carbonyls in SOA through the ozonolysis of ozone. The language needs improving throughout and the paper also needs significant restructuring and shortening. I think if these edits were made, it would be much easier for the reader to appreciate the significance of the results. The figures are clear and give the reader a flavour, but the text needs substantial improvement.

The introduction is far too long. Also, significant quantities of introductory material appear in the results (lines 264-285 and 325-340). This material should all appear in the intro, which itself needs to be written more concisely. The authors should also try and use better paragraph structure in places and use shorter paragraphs in general.

Another general point is that the methods do not provide enough detail. For each technique discussed, the authors should provide detection limits, precision and accuracy. Throughout, there is a tendency to talk about things being higher, lower or different. It would be better if such expressions were quantified where possible.

Finally, the experiments are carried out under conditions far removed from most ambient atmospheres, both in terms of concentrations of limonene and ozone, but also temperature (277K). Given this, I was missing a statement of the applicability of these results to real atmospheric conditions, or indeed as alluded to in the manuscript, indoor environments.

Specific comments:

1. I am not going to list every grammatical error, but I will mention a few here. I think that when the authors talk about 'SOA property' they mean SOA composition. This phrase should be replaced throughout with something more informative. Similarly, I am not clear what they mean by 'oxidants transition'. Again, clarification is needed.

- 2. 'Particulate unstable peroxides' is an awkward expression.
- 3. Line 64 'eaters' esters?

4. Line 542: Assume these are Kp coefficients - should specify. I don't understand the explanation for the finding that Kp is much bigger than estimated, line 556 (and lines 579-580). The language needs to be improved. What impact could experimental conditions have (low T and high precursor concentrations).

---

## Referee Comment (RC2) · Anonymous Referee #2 · 15 Jun 2018

Summary "Understanding the oxidants transition and SOA property in limonene ozonolysis: Role of different double bonds, radical chemistry, and water" describes a range of aspects of SOA formation from limonene ozonolysis. Experiments were carried out in a flow tube reactor with varying oxidant to limonene ratios and humidity. Scavengers were used to reduce OH concentrations during oxidation. Measurements were based on filter-based methods and collection of gas phase compounds in a coil collector. Peroxides were formed in significant amounts that increased with higher ozone to limonene ratio, suggesting the importance of the ozonolysis of the exo double bond in their formation. The formation and partitioning of carbonyls was also determined, and the partition coefficients (Kp) were found to be orders of magnitude larger

than predicted. The sustained production of H2O2 from sampled particles over the course of several days suggests varying ability of the limonene ozonolysis products to form oxidants in aerosol. Humidity was shown to consistently increase the formation of products that could evolve hydrogen peroxide in the particle phase. Small peroxide formation (e.g. peroxyformic acid) had increased production with higher RH. There were small effects of humidity on formation of larger peroxides in particles and a general decrease in gas-phase larger peroxide formation with increasing RH. The stability of peroxides formed was estimated based on total peroxides and hydrogen peroxide evolution after SOA formation. This manuscript present a number of interesting results and could be published with significant improvement in clarity and presentation.

General Comments:

It seems that "SOA property" should be replaced by SOA composition.

It seems that "oxidation transition" should be replaced by "oxidation regime" or something similar. Do you mean to indicate the effect of adding OH scavenger? Please clarify.

There are a number of places where introductory material shows up in the results and discussion, such as the historical perspective on OH formation in section 3.3. Please move this material to the introduction. The introduction itself needs a much clearer/more logical flow of ideas.

Add a clearer discussion/explanation of why the two OH scavengers had different effects. Add a clearer discussion of measurement techniques, particular the coil collector, with diagrams of the flow reactor and coil collector in the supplement.

The differences in yields are often overstated or exaggerated. For example, the peroxide mass fraction was determined in the three scavenger cases and discussed as if there were significant differences among these cases (Lines 484-487). But in fact, the differences were fairly small. Please check over all comparisons in the manuscript

and only state differences if they can be supported statistically, otherwise characterize them as similar.

A theme of the paper is the different effects from each of the two double bonds in limonene reacting with ozone. But your discussion of the ozonolysis of the endo and exo double bonds (e.g, Lines 320-323) is misleading, because the ozonolysis of the endo-DB will generally precede ozonolysis of the exo-DB. Your experiments can only isolate the ozonolysis of the endo-DB due to the discrepancy in reaction rates, which you do state. The ozonolysis of what was originally the exo-DB may not occur on the exo-DB of limonene, but rather on the remaining DB in an ozonolysis product of limonene. This is clearly stated in the final paragraph of Herrmann et al. "It should be noted that the measured OH-radical yield of the second double bond is the sum of all possible products formed from the reaction of ozone with the first double bond of the monoterpene."(1)

Too often the past tense is used incorrectly. For example, line 288 should read "Figure 3 shows. . .".

Calculated values are rarely supported by equations that clearly show how the calculation was done (however simple they may be). Add equations, either in the main text or supplement, that show how all yields are calculated.

Specific Comments

Figures 1 and 2. Neither of these needs to be in the main manuscript, both should be moved to supplemental. You should replace these with a scheme that indicates what chemistry you are investigating, particularly to show that the second DB oxidized is part of the products from the initial ozonolysis (such as Fig.2 from Herrmann et al.).(1)

Figure 3. The different markers are hard to distinguish. Please change the markers you use to make it easier to determine what experiment each line corresponds to. You might also label the two groups of lines (low and high ratios of ozone to limonene).

Figure 6. The lines are not helpful, please use just markers for each data point.

Lines 75-79 The ability of SCI to react with water is grossly overstated, even though it may be an important source of peroxides in the case of limonene. Review the discussions and add references to publications such as Long et al.(2) and Drozd et al.(3) that discuss lifetimes of SCI for bimolecular reaction and unimolecular decomposition.

Line 265-267 Specify the lifetimes for each loss process. You need to define "relatively long lifetime" with current quantitative estimates and relate this to unimolecular decomposition.

Line 310 What reactions were suppressed? Support or contrast this result with literature on SCI reactivity.

Line 317. You need to add equations that show how you calculate these yields, possibly in the supplemental.

Section 3.5.3 You need to show in an equation how you calculate the relative amounts of stable and unstable peroxides.

Line 479 Why was a MW of 300 g/mol assumed? Support with references or appropriate justification.

Line 504 Awkward sentence. What does "chemical effect" mean in this case?

There are many grammatical errors and awkward sentences. Below is a partial list of these. Check over the entire manuscript again.

13 – "radical chemistry" 25 "Considerable generation of H2O2 from SOA in the aqueous phase" 28 "SOA composition" (and throughout paper) 39-40 "suggest that the...need further study" 45 "Total monoterpene emissions are estimated to be.." 47 "non-negligible..." 64 "esters" 79 "reactions of alkenes..." 103 "Due to abundant water vapor..." 169 "first generation" 222 "constituents were generated" 280 "prior to generating aldehyde" 293 "There was little to no effect of scavenger" 385 "chamber studies" 516

"HACE might be generated. . ."

References

1 F. Herrmann, R. Winterhalter, G.K. Moortgat, and J. Williams, Atmos. Environ. 44, 3458 (2010). 2 B. Long, J.L. Bao, and D.G. Truhlar, Proc. Natl. Acad. Sci. 115, 6135 (2018). 3 G.T. Drozd, T. Kurtén, N.M. Donahue, and M.I. Lester, J. Phys. Chem. A 121, 6036 (2017).

---

## Author Comment (AC1) · 2 Aug 2018

**Response to Reviewer 1**

We gratefully thank you for your constructive comments and thorough review. Below are our responses to your comments.

(Q=Question, and A=Answer)

**General Comments:**

Q1. The introduction is far too long. Also, significant quantities of introductory material appear in the results (lines 264-285 and 325-340). This material should all appear in the intro, which itself needs to be written more concisely. The authors should also try and use better paragraph structure in places and use shorter paragraphs in general.

A: Thanks for your suggestion. All of the introductory materials shown in the Sect. 3.2, Sect. 3.3, and Sect. 3.4 are now moved to the introduction. The introduction has been restructured and shortened.

Q2. Another general point is that the methods do not provide enough detail. For each technique discussed, the authors should provide detection limits, precision and accuracy.

A: We follow the advice of the reviewer. The detection limits, precision, and accuracy have been provided for the methods of detecting peroxides and carbonyls in the revised text. We also provide the diagrams of the flow tube reactor, the coil collector, and the Horibe tube in the Supplement

Q3. Throughout, there is a tendency to talk about things being higher, lower or different. It would be better if such expressions were quantified where possible.

A: Thanks for your suggestion. We have checked our manuscript and changed such expressions.

Q4. Finally, the experiments are carried out under conditions far removed from most ambient atmospheres, both in terms of concentrations of limonene and ozone, but also temperature (277K). Given this, I was missing a statement of the applicability of these results to real atmospheric conditions, or indeed as alluded to in the

manuscript, indoor environments.

A: We regret that we did not clarify well enough. All of our experiments were conducted at 298 K as stated in the Sect. 2.2. The temperature of 277 K was just used to keep the SOA solution in order to maintain the stability of samples and prolong their storage time. We have explained this in the revised manuscript. As for the concentrations of reactants used in our experiments, the concerns of the reviewer are reasonable. To get enough products for analysis in a short reaction time, both of the concentrations of limonene and ozone in this study were obviously higher than those in the real atmospheric conditions, which might have influence on the gas-phase and particle-phase chemistry. So the effects of the reactants concentrations on the experimental results are discussed below and this part is now added in the Supplement.

A major impact of the high concentrations of reactants is the increased $RO_2$ concentration. In recent years many studies reported that the autoxidation processes formed highly oxidized $RO_2$ radicals, which reacted with $HO_2$ and other $RO_2$ radicals forming highly oxidized multifunctional organic compounds (HOMs) (Jokinen et al., 2014; Richters et al., 2016a, b). The production of HOMs is controlled by two competing processes, i.e., $RO_2$ autoxidation vs. $RO_2$ reaction with $HO_2$ and other $RO_2$ radicals. Zhang et al. (2015) found that at low α-pinene levels, the longer lifetime of $RO_2$ radicals favored the isomerization pathways and consequently led to enhanced ELVOC dimers production. They estimated that the corresponding lifetime of $RO_2$ radicals decreased by less than an order of magnitude when the initial α-pinene mixing ratio increased from 10 ppbv to 150 ppbv, which was not sufficient to perturb the dynamics of overall $RO_2$ chemistry. In our experiments, where the limonene concentration was below 200 ppbv, we speculated that although the $RO_2$ chemistry was affected to some extent it would not bring huge influence on the results. When the $RO_2$ concentration is high, the reactions of SCIs and $RO_2$ radicals might happen in the system (Sadezky et al., 2008). Zhao et al. (2015) found that the reactions of SCIs and $RO_2$ radicals played a key role in particle formation in *trans*-3-hexene ozonolysis, while for large

alkenes such as terpenes and sesquiterpenes such reactions might be unimportant. Thus, although the concentrations of reactants used in this study were higher than that in the real atmospheric condition, the SCIs + $RO_2$ reactions did not have a huge effect on the reaction system. The SOA yield of limonene ozonolysis observed in this study was in the range of the values reported before (Ahmad et al., 2017; Chen and Hopke, 2010; Pathak et al., 2012). It is true that when the mixing ratios of reactants are high, the gas-particle partitioning processes of semi-volatility and low-volatility products are promoted resulting in higher SOA yield, yet we think that it may not have great impact on the representativeness of the products we investigated in particles.

**Specific Comments:**

Q6. I am not going to list every grammatical error, but I will mention a few here. I think that when the authors talk about 'SOA property' they mean SOA composition. This phrase should be replaced throughout with something more informative. Similarly, I am not clear what they mean by 'oxidants transition'. Again, clarification is needed.

A: Thanks for your suggestion and we have changed "SOA property" to "SOA composition" throughout the text. Besides, "oxidants transition" is replaced by "oxidation regime" in the revised manuscript. In this study, the objective of investigating the oxidation regime of alkene ozonolysis is to understand the formation of the oxidizing products in limonene ozonolysis. These compounds, including OH radicals, stabilized Criegee intermediates, and peroxides, are critical to atmospheric oxidation processes since they own the power of oxidizing other species. We clarify this in the revised introduction.

Q7. 'Particulate unstable peroxides' is an awkward expression.

A: We have changed that to "unstable peroxides in particles".

Q8. Line 64 – 'eaters' - esters?

A: Yes, we have revised it.

Q9. Line 542: Assume these are Kp coefficients - should specify. I don't understand the explanation for the finding that Kp is much bigger than estimated, line 556 (and lines 579-580). The language needs to be improved. What impact could experimental conditions have (low T and high precursor concentrations).

A: We have specified that these are the calculated gas-particle partitioning coefficients. As for the explanation for the finding that the measured $K_p$ is much bigger than the predicted $K_p$, we have clarified it better in the revised text. A plausible explanation for the large difference between the measured $K_p$ and the predicted $K_p$ was that carbonyl compounds were easy to polymerize and react with other species on particles, resulting in that these carbonyls existed in forms of hydrates and oligomers (Corrigan et al., 2008; Hastings et al., 2005; Kroll et al., 2005; Volkamer et al., 2007). The hydrates and oligomers of carbonyls have much lower vapor pressures than their precursors, and they could reversibly return to their carbonyl monomers during analysis (Healy et al., 2008; Ortiz et al., 2013; Toda et al., 2014). The temperature used here was 298 K and the effect of precursor concentrations was analyzed. When the mixing ratios of reactants are high, the gas-particle partitioning processes of semi-volatility and low-volatility products are promoted resulting in higher SOA yield. However, when we calculated the gas-particle partitioning coefficients, the effect of the concentration of total suspended particulate matter was taken into consideration as shown by Eq. (2), so the higher SOA concentration caused by higher precursor concentrations would not impact the partitioning behaviors of carbonyls. Another effect of the high precursor concentrations was that it might promote the reactions of carbonyl compounds on particles, yet the concentrations of most carbonyls detected in our reactions were usually several ppbv, which were just a little higher than those in the real atmosphere, so we speculated that this impact on the partitioning coefficients was also limited.

**References**

Ahmad, W., Coeur, C., Cuisset, A., Coddeville, P., and Tomas, A.: Effects of scavengers of Criegee intermediates and OH radicals on the formation of secondary organic aerosol in the ozonolysis of limonene, J. Aerosol Sci., 110, 70–83, doi: 10.1016/j.jaerosci.2017.05.010, 2017.

Chen, X., and Hopke, P. K.: A chamber study of secondary organic aerosol formation by limonene ozonolysis, Indoor Air, 20, 320–328, doi: 10.1111/j.1600-0668.2010.00656.x, 2010.

Corrigan, A. L., Hanley, S. W., and Hann, D. O. D.: Uptake of glyoxal by organic and inorganic aerosol, Environ. Sci. Technol., 42, 4428–4433, doi: 10.1021/es7032394, 2008.

Hastings, W. P., Koehler, C. A., Bailey, E. L., and Haan, D. O. D.: Secondary organic aerosol formation by glyoxal hydration and oligomer formation: Humidity effects and equilibrium shifts during analysis, Environ. Sci. Technol., 39, 8728–8735, doi: 10.1021/es050446l, 2005.

Healy, R. M., Wenger, J. C., Metzger, A., Duplissy, J., Kalberer, M., and Dommen, J.: Gas/particle partitioning of carbonyls in the photooxidation of isoprene and 1,3,5-trimethylbenzene, Atmos. Chem. Phys., 8, 3215–3230, doi: 10.5194/acp-8-3215-2008, 2008.

Jokinen, T., Sipilä, M., Richters, S., Kerminen, V. M., Paasonen, P., Stratmann, F., Worsnop, D., Kulmala, M., Ehn, M., Herrmann, H., and Berndt, T.: Rapid autoxidation forms highly oxidized $RO_2$ radicals in the atmosphere, Angew. Chem. Int. Ed., 53, 14596–14600, doi: 10.1002/anie.201408566, 2014.

Kroll, J. H., Ng, N. L., Murphy, S. M., Varutbangkul, V., Flagan, R. C., and Seinfeld, J. H.: Chamber studies of secondary organic aerosol growth by reactive uptake of simple carbonyl compounds, J. Geophys. Res., 110, D23207, doi: 10.1029/2005JD006004, 2005.

Ortiz, R., Shimada, S., Sekiguchi, K., Wang, Q., and Sakamoto, K.: Measurements of changes in the atmospheric partitioning of bifunctional carbonyls near a road in a suburban area, Atmos. Environ., 81, 554–560,

doi: 10.1016/j.atmosenv.2013.09.045, 2013.

Pathak, R. K., Salo, K., Emanuelsson, E. U., Cai, C., Lutz, A., Hallquist, A. M., and Hallquist, M.: Influence of ozone and radical chemistry on limonene organic aerosol production and thermal characteristics, Environ. Sci. Technol., 46, 11660–11669, doi: 10.1021/es301750r, 2012.

Richters, S., Herrmann, H., and Berndt, T.: Different pathways of the formation of highly oxidized multifunctional organic compounds (HOMs) from the gas-phase ozonolysis of β-caryophyllene, Atmos. Chem. Phys., 16, 9831−9845, doi: 10.5194/acp-16-9831-2016, 2016a.

Richters, S., Herrmann, H., and Berndt, T.: Highly oxidized $RO_2$ radicals and consecutive products from the ozonolysis of three-sesquiterpenes, Environ. Sci. Technol., Environ. Sci. Technol., 50, 2354−2362, doi: 10.1021/acs.est.5b05321, 2016b.

Sadezky, A., Winterhalter, R., Kanawati, B., Römpp, A., Spengler, B., Mellouki, A., Le Bras, G., Chaimbault, P., and Moortgat, G. K.: Oligomer formation during gas-phase ozonolysis of small alkenes and enol ethers: new evidence for the central role of the Criegee intermediate as oligomer chain unit, Atmos. Chem. Phys., 8, 2667–2699, doi:10.5194/acp-8-2667-2008, 2008.

Toda, K., Yunoki, S., Yanaga, A., Takeuchi, M., Ohira, S., Dasgupta, P. K., Formaldehyde content of atmospheric aerosol, Environ. Sci. Technol., 48, 6636–6643, doi: 10.1021/es500590e, 2014.

Volkamer, R., Martini, F. S., Molina, L. T., Salcedo, D., Jimenez, J. L., Molina, M. J.: A missing sink for gas-phase glyoxal in Mexico City: Formation of secondary organic aerosol, Geophys. Res. Lett., 34, L19807, doi: 10.1029/2007GL030752, 2007.

Zhao, Y., Wingen, L. M., Perraud, V., Greaves, J., and Finlayson-Pitts, B. J.: Role of the reaction of stabilized Criegee intermediates with peroxy radicals in particle formation and growth in air, Phys. Chem. Chem. Phys., 17, 12500–12514, doi:10.1039/C5CP01171J, 2015.

Zhang, X., McVay, R. C., Huang, D. D., Dalleska, N. F., Aumont, B., Flagan, R. C., and Seinfeld, J. H.: Formation and evolution of molecular products in α-pinene secondary organic aerosol, Proc. Natl. Acad. Sci., 112, 14168–14173, doi: 10.1073/pnas.1517742112, 2015.

---

## Author Comment (AC2) · 2 Aug 2018

**Response to Reviewer 2**

We gratefully thank you for your constructive comments and thorough review. Below are our responses to your comments.

(Q=Question, and A=Answer)

**General Comments:**

Q1. It seems that "SOA property" should be replaced by SOA composition.

A: We have revised it.

Q2. It seems that "oxidation transition" should be replaced by "oxidation regime" or something similar. Do you mean to indicate the effect of adding OH scavenger? Please clarify.

A: Thanks for your suggestion and "oxidation regime" is used in the revised manuscript. In this study, the objective of investigating the oxidation regime of alkene ozonolysis is to understand the formation of the oxidizing products in limonene ozonolysis. These compounds, including OH radicals, stabilized Criegee intermediates, and peroxides, are critical to atmospheric oxidation processes since they own the power of oxidizing other species. We clarify this in the revised introduction.

Q3. There are a number of places where introductory material shows up in the results and discussion, such as the historical perspective on OH formation in section 3.3. Please move this material to the introduction. The introduction itself needs a much clearer/more logical flow of ideas.

A: We follow the advice of the reviewer and all of the introductory materials shown in the Sect. 3.2, Sect. 3.3, and Sect. 3.4 are now moved to the introduction. The introduction has been restructured and shortened.

Q4. Add a clearer discussion/explanation of why the two OH scavengers had different effects.

A: The reviewer concerns the effect of two OH scavengers hence the following content has been added into the Sect. 3.5.4 of the revised text.

Researches before have provided evidence for OH radicals formation in ozonolysis experiments and OH scavenger is often used to avoid the disturbance of OH reaction. 2-butanol and cyclohexane used here are both commonly used OH scavengers, and Chew and Atkinson (1996) showed that there was no difference in their abilities to scavenge OH radicals. However, it should be noted that OH scavenger could convert OH radicals into a mixture of hydroperoxy ($HO_2$) and alkylperoxy ($RO_2$) radicals and higher $[HO_2]/[RO_2]$ is observed when 2-butanol is used (Docherty and Ziemann, 2003; Jonsson et al., 2008). As shown in the Sect. 3.6.1, the fact that both at low and high $[O_3]/[limonene]$, the SOA yield was higher with 2-butanol than with cyclohexane suggested that the increase of $HO_2$ concentration promoted the SOA formation. This result is consistent with that suggested by Keywood et al. (2004) who observed the higher $[HO_2]/[RO_2]$ resulting in higher SOA yield in cyclohexene ozonolysis, while Docherty and Ziemann (2003) showed that increased $[HO_2]/[RO_2]$ inhibited aerosol formation in β-pinene ozonolysis.

Some studies demonstrated that acid and peroxide products were sensitive to OH scavengers (Ahmad et al., 2017; Henry and Donahue, 2011; Ma et al., 2008). Here through investigating the stabilities of particulate peroxides formed in different conditions, we showed that the choice of OH scavenger would influence the types of particulate peroxides produced in the reactions. In the Sect. 3.5.3, we suggest that peroxycarboxylic acids and peroxyhemiacetals be the main components of unstable peroxides in particles. The peroxyhemiacetals formed by heterogeneous reactions of peroxides and aldehydes could dissociate into these species, yet the formation of peroxyhemiacetals was not supposed to be affected by OH scavenger since both of the peroxides and carbonyl compounds observed in the gas phase did not show large differences when OH scavenger changed. We speculate that the impact of OH scavenger on particulate peroxides could be mainly attributed to the formation of peroxycarboxylic acids under different $[HO_2]/[RO_2]$. Keywood et al. (2004) indicated that the OH scavenger impacted the $HO_2$ reaction with acylperoxy radicals forming acid and peracid products, which were considered

to have low volatility. With higher $HO_2$ concentration, the SOA yield in the presence of 2-butanol was higher than that in the presence of cyclohexane, since the $HO_2$-acylperoxy reactions contributed to some low-volatility products. However, the $RO_2$ radicals formed from cyclohexane may also participate into reactions and help produce more unstable peroxycarboxylic acids which could partition into the particle phase.

Q5. Add a clearer discussion of measurement techniques, particular the coil collector, with diagrams of the flow reactor and coil collector in the supplement.

A: We have better explained the measurement techniques in the revised manuscript. The coil collector is around 30 cm long and its effective length is about 100 cm. The coil is similar with that used in earlier studies (Grossmann et al., 2003; Sauer et al., 1996, 1997) and is controlled at a temperature of 4 °C with $H_3PO_4$ solution (pH 3.5) serving as the rinsing solution. The diagrams of the flow tube reactor, the coil collector, and the Horibe tube are added in the Supplement.

Q6. The differences in yields are often overstated or exaggerated. For example, the peroxide mass fraction was determined in the three scavenger cases and discussed as if there were significant differences among these cases (Lines 484-487). But in fact, the differences were fairly small. Please check over all comparisons in the manuscript and only state differences if they can be supported statistically, otherwise characterize them as similar.

A: We have revised it and other comparisons in the text are also checked.

Q7. A theme of the paper is the different effects from each of the two double bonds in limonene reacting with ozone. But your discussion of the ozonolysis of the endo and exo double bonds (e.g, Lines 320-323) is misleading, because the ozonolysis of the endo-DB will generally precede ozonolysis of the exo-DB. Your experiments can only isolate the ozonolysis of the endo-DB due to the discrepancy in reaction rates, which you do state. The ozonolysis of what was originally the exo-DB may not occur on the exo-DB of limonene, but

rather on the remaining DB in an ozonolysis product of limonene. This is clearly stated in the final paragraph of Herrmann et al. "It should be noted that the measured OH-radical yield of the second double bond is the sum of all possible products formed from the reaction of ozone with the first double bond of the monoterpene."[1]

A: We regret that in the previous version we did not clarify well enough. We have added an explanation in Sect. 3.2 for that. The SCIs yield of exocyclic DB ozonolysis estimated here was not for a specific product formed from endocyclic DB ozonolysis, but the sum of first-generation products with a remaining double bond.

Q8. Too often the past tense is used incorrectly. For example, line 288 should read, "Figure 3 shows . . ."

A: We have revised it and other similar mistakes are also corrected.

Q9. Calculated values are rarely supported by equations that clearly show how the calculation was done (however simple they may be). Add equations, either in the main text or supplement, that show how all yields are calculated.

A: We follow the advice of the reviewer and in the Supplement a summary of the calculation equations have been added, which includes the equations to calculate the wall loss fraction, the molar yield of products, the molar yield of SCIs, the SOA yield, and the mass fraction of particulate peroxides.

**Specific Comments:**

Q10. Figures 1 and 2. Neither of these needs to be in the main manuscript, both should be moved to supplemental. You should replace these with a scheme that indicates what chemistry you are investigating, particularly to show that the second DB oxidized is part of the products from the initial ozonolysis (such as Fig.2 from Herrmann et al.). [1]

A: Figure 1 and Figure 2 have been moved to the Supplement and a scheme of endocyclic and exocyclic double bonds of limonene reaction with ozone (Figure 2) is added in the revised manuscript.

Q11. Figure 3. The different markers are hard to distinguish. Please change the markers you use to make it

easier to determine what experiment each line corresponds to. You might also label the two groups of lines (low and high ratios of ozone to limonene).

A: Thanks for your suggestion. We have changed the markers and labeled the two groups of lines in Figure 1 and Figure 3 in the revised text.

Q12. Figure 6. The lines are not helpful, please use just markers for each data point.

A: We have revised it.

Q13. Lines 75-79 The ability of SCI to react with water is grossly overstated, even though it may be an important source of peroxides in the case of limonene. Review the discussions and add references to publications such as Long et al.(2) and Drozd et al.(3) that discuss lifetimes of SCI for bimolecular reaction and unimolecular decomposition.

A: We thank the reviewer for pointing us to these references, which we now include in the revised paper. Besides, we have clarified in the Sect. 3.2 that considering that a portion of SCIs might undergo unimolecular decomposition, the SCIs yield estimated here represented a lower limit. In the Sect. 3.3 the results demonstrated that the OH yield was not obviously affected by RH, we speculated that the fraction of SCIs that decomposed and formed OH radicals was small.

Q14. Line 265-267 Specify the lifetimes for each loss process. You need to define "relatively long lifetime" with current quantitative estimates and relate this to unimolecular decomposition.

A: Thanks for your suggestion and we have defined the lifetimes of SCIs in the Sect. 3.2. In previous studies, the unimolecular decomposition of SCIs and the reactions of SCIs with water showed strong structure dependence. For $CH_2OO$ and anti-$CH_3CHOO$, the atmospheric lifetimes of their bimolecular reactions with $H_2O$ and $(H_2O)_2$ were less than 1 ms, while their lifetimes of unimolecular reaction were much longer (Lin et al., 2016; Sheps et al., 2014; Taatjes et al., 2013; Welz et al., 2012). For syn-$CH_3CHOO$ and $(CH_3)_2COO$, the

atmospheric lifetimes of their bimolecular reactions with $H_2O$ and $(H_2O)_2$ were more than 100 ms, while their lifetimes of unimolecular reaction were just few milliseconds at 298 K (Drozd et al., 2017; Huang et al., 2015; Long et al., 2018; Sheps et al., 2014; Taatjes et al., 2013; Welz et al., 2014). For larger SCIs, their atmospheric lifetimes of unimolecular reaction and bimolecular reactions with $H_2O$ and $(H_2O)_2$ were not quantified. Tillmann et al. (2010) inferred that about 46% ECIs formed from α-pinene ozonolysis would stabilize and at 44% RH about 65% SCIs formed $H_2O_2$ after reacting with water and 28% SCIs decomposed, Yao et al. (2014) showed that a large fraction of ECIs formed from α-cedrene were stabilized and the unimolecular decomposition of SCIs was suppressed by their bimolecular reactions. Although we could not estimate the rates of SCIs decomposition and their reaction with water, the results here proved that reaction with water was an essential route for limonene SCIs, and the rapid decomposition of HAHPs made an important contribution to $H_2O_2$ formation. At high RH, the unimolecular decomposition of limonene SCIs could be suppressed by their bimolecular reactions with water monomer and water dimer.

Q15. Line 310 What reactions were suppressed? Support or contrast this result with literature on SCI reactivity.

A: We regret for the unclear expression and we have better explained this in the revised manuscript. These suppressed reaction channels include the unimolecular decomposition of SCIs and the reactions of SCIs with other products in the system. For SCIs containing three or less carbon atoms, their unimolecular decomposition and reactions with water showed strong structure dependence, which have been clarified in the revised text. However, as for large SCIs their reaction pathways are still not clear. Tillmann et al. (2010) inferred that about 46% ECIs formed from α-pinene ozonolysis would stabilize and at 44% RH about 65% SCIs formed $H_2O_2$ after reacting with water and 28% SCIs decomposed, Yao et al. (2014) showed that a large fraction of ECIs formed from α-cedrene were stabilized and the unimolecular decomposition of SCIs was suppressed by their bimolecular reactions. The results in this study indicated that at high RH the unimolecular decomposition of

limonene SCIs could be suppressed by their bimolecular reactions with water monomer and water dimer.

Q16. Line 317. You need to add equations that show how you calculate these yields, possibly in the supplemental.

A: We have added equations in the Supplement.

Q17. Section 3.5.3 You need to show in an equation how you calculate the relative amounts of stable and unstable peroxides.

A: We have added the equation in Sect. 3.5.3.

Q18. Line 479 Why was a MW of 300 g/mol assumed? Support with references or appropriate justification.

A: The average molecular weight of peroxides in particles is assumed to be 300 g/mol, which is estimated to be slightly less than the molecular weight of peroxyhemiacetals, and this value has been used to calculate the mass of particulate peroxides in some studies (Docherty et al., 2005; Nguyen et al., 2010; Surratt et al., 2006). We clarify this in the revised manuscript.

Q19. Line 504 Awkward sentence. What does "chemical effect" mean in this case?

A: We are sorry for not explaining that clearly. The chemical effect means the effect of water participating in some gas-phase and particle-phase reactions resulting in more low-volatility peroxides formation. This is now better explained in the revised text.

Q20. There are many grammatical errors and awkward sentences. Below is a partial list of these. Check over the entire manuscript again.

13 – "radical pha chemistry" 25 "Considerable generation of H2O2 from SOA in the aqueous se" 28 "SOA composition" (and throughout paper) 39-40 "suggest that the. . .need further study" 45 "Total monoterpene emissions are estimated to be.." 47 "non-negligible. . ." 64 "esters" 79 "reactions of alkenes. . ." 103 "Due to abundant water vapor..." 169 "first generation" 222 "constituents were generated" 280 "prior to generating

aldehyde" 293 "There was little to no effect of scavenger" 385 "chamber studies" 516"HACE might be generated. . ."

A: Thanks for your suggestion and we have revised all of these mistakes mentioned above. We also check over the manuscript seriously and correct other errors.

**References**

[revised manuscript text omitted]

---

## Author Response (AR2)

**ACP Editor**

Dear Prof. Markus Ammann,

Enclosed please find our revised manuscript entitled "The oxidation regime and SOA composition in limonene ozonolysis: Roles of different double bonds, radicals, and water", revised supplement and the response to the comments by you and the two anonymous referees. We have corrected the technical errors and the English. A native English speaker has helped us to polish the English language.

Detailed changes made in the manuscript can be seen in the marked-up version in this response.

Thanks for your time.

Sincerely yours,

Zhongming Chen and co-authors

**Response to Referee 1**

We gratefully thank the reviewer for the comments and we have polished the language and checked the grammar. Below are our responses to the comments and the responses are in blue. Other corrections are shown in the revised manuscript.

28: specify the 0.2 and 0.4-0.6 are yields

We have clarified in the abstract that these numbers represent the mass fraction of particulate peroxides in SOA.

69-70: add references

We add the relevant references in the revised text.

87: needs

Yes, we have revised it.

95: include more recent references here

We have added more recent references in the revised manuscript.

97: have paid

Yes, we have revised it.

99-101: rewrite without the phrases "on the one hand" and "on the other hand"

We follow the advice of the reviewer and the sentence is rewritten.

151-152: processes

Yes, we have revised it.

162: we take the peroxides.... to be HMW peroxides

Yes, we have revised it.

241: discussed?

We have changed that to "investigated".

251-252 (and throughout paper): Do not use the term "molar number" use number of moles Thanks for your suggestion and we have changed such expressions throughout the paper.

275: is slow

Yes, we have revised it.

297-298: remove this sentence

Yes, we have removed the sentence.

301: generated

Yes, we have revised it.

303 (and throughout): change to "SCI yield"

Yes, we have changed that throughout the paper.

317: was generated

Yes, we have revised it.

320: yield the same as

Yes, we have revised it.

322: show a clear dependence

Yes, we have revised it.

346: PFA and PAA were mainly formed through ECI isomerization

Yes, we have revised it.

354: "similar formation mechanisms"

Yes, we have revised it.

361: radical generation

Yes, we have revised it.

379: inclined to have

Yes, we have revised it.

394: remove concretely

Yes, we have removed it.

398: for six days

Yes, we have revised it.

401: under dry conditions

Yes, we have revised it.

414-416: change to number of moles

Yes, we have revised it.

423: may have hydrolyzed to form

Yes, we have revised it.

455: "not supposed to be.." I believe you mean "was not considered to be"

Yes, we have revised it.

490: we are the first to report

Yes, we have revised it.

500: degree of oxidation

Yes, we have revised it.

503: the effect of water

Yes, we have revised it.

508-511: rewrite without the phrases "on the one hand" and "on the other hand"

We follow the advice of the reviewer and the sentence is rewritten.

552: Figure 6 shows

Yes, we have revised it.

564: oxidizing products including

Yes, we have revised it.

583: Limonene is unique in many aspects because of its different DBs, suggesting that terpenes with multiple double bonds may have more complex effects on the atmosphere than previously thought.

We follow the advice of the reviewer and the sentence is rewritten.

585 degree of oxidation

Yes, we have revised it.

**Response to Referee 2**

We gratefully thank the reviewer for the comments we have polished the language and checked the grammar. Below are our responses to the comments and the responses are in blue. Other corrections are shown in the revised manuscript.

1. Lines 38-40: grammar

We have rewritten that sentence in the revised manuscript.

2. Line 58: isomers

Yes, we have revised it.

3. Line 82: references needed

We add the relevant references in the revised text.

4. Line 282: quantify

Yes, we have quantified the precision.

5. Line 448-449: grammar

We have rewritten the sentence in the revised manuscript.

6. Line 518-520: grammar

We have rewritten the sentence in the revised manuscript.

7. Line 556: grammar

We have rewritten the sentence in the revised manuscript.

[revised manuscript text omitted]
 (NH3·H2O, Beijing Tongguang Fine Chemicals Company, ≥ 99.5%), sulfuric acid (H2SO4, Xilong Chemical Company, 95.0–

125 98.0%), ultrapure water (18MΩ, Millipore), N2 (≥ 99.999%, Beijing Haikeyuanchuang Practical Gas Company Limited, Beijing, China), O2 (≥ 99.999%, Beijing Haikeyuanchuang Practical Gas Company Limited, Beijing, China), polytetrafluoroethylene (PTFE) filter membrane (Whatman Inc., 47\_mm in diameter), and quartz microfiber filters (Whatman Inc.)-were used in this study.

**2.2 Apparatus and procedures**

- 130 A flow tube reactor (2 m length, 70 mm inner diameter, quartz wall)2 equipped with a water jacket for controlling the temperature2 was used to investigate the ozonolysis of limonene. All the experiments were conducted at 298±0.5 K in darknessthe-dark. O3 was generated by O2 photolysis in a 2 L quartz tube with a low-pressure Hg lamp, and the detailed quantification method of O3 was described in our previous study (Chen et al., 2008). H2O2 produced by UV irradiation of O2 and trace water was measured in control experiments and
- deducted from the results. Limonene gas was generated by passing N2 flow over liquid limonene in a diffusion
  tube at the selected temperature2 and OH scavenger (2-butanol or cyclohexane) gas was generated with a
  bubbler. The concentrations of limonene and the OH scavenger were determined by gas chromatography with a
  flame ionization detector (GC-FID, Agilent 7890A, USA). Water vapor was generated by passing N2 through a
  water bubbler, which contained a carborundum disc submergedsubmerging in ultrapure water (18 MΩ). The
  mixing gas2 including limonene, OH scavenger, ozone, and dry or wet synthetic air (80% N2 and 20% O2), was
  successively introduced into the reactor2, and with With a total flow rate of 2 standard L min-1, the residence
  time was estimated to be 240 s.

To explore the reaction mechanism of endocyclic and exocyclic DBs ozonolysis; and the effect of multi-generation oxidation in limonene ozonolysis, we conducted two sets of experiments with different ratios of ozone to limonene concentrationwere-conducted. In the following-content, [O3] denotesdenoted the concentration of ozone, [limonene] denotesdenoted the concentration of limonene, and [O3]/[limonene] denotesdenoted the concentration. In the low [O3]/[limonene]-set of experiments, the initial concentrations of limonene and ozone were ~280 ppbv and ~500 ppbv, respectively-In3 whereas in the high [O3]/[limonene]-set of experiments, the initial concentrations of limonene and ozone were ~

145

cyclohexane were added to scavenge OH radicals in the RH range of 0%–90%. In the tables and figures, the low-ratio and high-ratio-ratio sets of experiments arewere denoted by with marks L and H, respectively, and the conditions in the absence of scavenger, in the presence of 2-butanol, and in the presence of cyclohexane are denoted were represented by No-sca, 2-But, and C-hex, respectively. The experimental Experimental conditions arewere listed in Table 1.

According to previous studies, in limonene ozonolysis the rate constant of the endocyclic DB reaction with ozone in limonene ozonolysis iswas  $2 \times 10^{-16}$  cm3 molecule-1 s-1 (Atkinson, 1990; Shu and Atkinson, 1994), while the exocyclic DB reaction with ozone iswas about 30 times slower than the endocyclic DB reaction (Zhang et al., 2006). Based on those rate constants, we estimated that at low [O3]/[limonene], less than 1% exocyclic DB iswas ozonated, so this situation mainly represents represented the first-generation oxidation. In this circumstance, because the ozone concentration iswas low, the OH reaction would impact the amount of limonene consumed by O3. In the presence of OH scavenger, ~-42% endocyclic DB reacted with O3, while in the absence of scavenger, ~-38% endocyclic DB reacted with O3. At high [O3]/[limonene], more than 99% endocyclic DB and about 51% exocyclic DB reacted with ozone, and since the ozone concentration in this situation was high, the OH effect on the ozonolysis was presumed to be unimportant. The latter condition, which contained multi-generation oxidation processprocesses, was more likely to occurhappen since the ratio of [O3] to [limonene] was similar to the ratio in the real atmosphere.

It should be noted that one advantage of flow tube reactor iswas that the wall would be in equilibrium with the gas phase after a stationary period, and according According to our observations, this process usually requiresneeded about 2 h. Thus, In order to stabilize the system and diminish the wall effect to the extentas much as possible, we ran the reactor was usually aged for 2 h prior to taking measurements. and And after the experiments, the reactor was rinsed-out with ultrapure water and blown-to dry with N2.

**2.3 Products analysis**

175

180

To better investigate the gas-particle partitioning of products formed in limonene ozonolysis, we analyzed the gas-phase and particle-phase products simultaneously. We measured total Total peroxides and a series of low-molecule-weight (LMW) peroxides, were measured here, and in In the following discussion we regarded those the peroxides that could be detected by high performance liquid chromatography (HPLC) to been LMW peroxides, while eonsidered took the peroxides undetermined by HPLC asto be high-molecule-weight (HMW) peroxides. For particle-phase peroxides detection, a PTFE filter was used for the SOA collection, and the mass of SOA was measured by semi-micro balance (Sartorius, Germany). Since the control experiment-results showed that a long-duration time collection led to the loss of some peroxides in particles, the collection time was controlled to be 3 h for each filter, and the accuracy of the particulate products analysis was discussed in the Supplement. Each loaded PTFE filter was extracted with 20 mL H3PO4 solution (pH 3.5) using a shaker (Shanghai Zhicheng ZWY 103D, China) at 180 rpm and 4 °C for 15 min3 and the extraction efficiency was confirmed in our previous work (Li et al., 2016). For gas-phase peroxides detection, the air samples through the

- confirmed in our previous work (Li et al., 2016). For gas-phase peroxides detection, the air samples through the filter were collected in a glass coil collector at a temperature of 4 °C with H3PO4 solution (pH 3.5) serving as the rinsing solution. The coil collector is around 30 cm long and its effective length is about 100 cm. The coil is similar with that used in earlier studies (Grossmann et al., 2003; Sauer et al., 1996, 1997)a and the diagram of the thermostatic coil collector is provided shown in the Supplement.
- The detection method for detecting theof peroxides was reported in our previous studies (Hua et al., 2008; Li et al., 2016), so and is only a briefly described description was given here. LMW peroxides were analyzed by HPLC (Agilent 1100, USA) coupled with post-column derivatization and fluorescence detection on line. Peroxides separated by column chromatography reacted with *p*-hydroxyphenylacetic acid (POPHA) under the catalysis of hemin forming POPHA dimers, and then the dimers were quantified by fluorescence detector. The

195 standard solution of peroxides was prepared and used for calibration in each measurement. The detection limit for theof gas-phase LMW peroxides was about 22 ppty, and in the particle phase the detection limit was around 0.068 ng/ug for H2O2. The accuracy of this method was estimated to be around 7% with and the precision of the measurement results was usually within 20%. The concentration of total peroxides ( $H_2O_2$ , ROOH, and ROOR') was determined by the iodometric spectrophotometric method, which is based on the reaction of peroxides and 200 iodide ions (Docherty et al., 2005; Mutzel et al., 2013). Briefly, excessive KI solution was added tointo samples purged of Q2, after After remaining staying 12 24 h in the darkdarkness 12–24 h for the derivatization, the produced I3- 
[revised manuscript text omitted]
     | $0.065 \pm 0.006$ | $0.101 \pm 0.009$ | $0.087 {\pm} 0.011$ | 0.401±0.016       | $0.502{\pm}0.008$   | $0.477 {\pm} 0.010$ |
| 10% RH    | $0.091 \pm 0.010$ | 0.124±0.013       | 0.113±0.009         | 0.436±0.009       | $0.534 {\pm} 0.009$ | $0.502{\pm}0.013$   |
| 30% RH    | 0.125±0.010       | $0.147 \pm 0.011$ | 0.143±0.015         | 0.458±0.020       | 0.553±0.015         | $0.506{\pm}0.011$   |
| 50% RH    | $0.149 \pm 0.007$ | $0.174 \pm 0.011$ | 0.161±0.016         | 0.466±0.016       | 0.571±0.009         | $0.512{\pm}0.007$   |
| 70% RH    | 0.155±0.009       | $0.178 \pm 0.009$ | 0.169±0.014         | 0.486±0.023       | 0.576±0.010         | $0.503{\pm}0.011$   |
| 80% RH    | 0.169±0.013       | 0.189±0.010       | 0.189±0.012         | 0.492±0.015       | 0.580±0.013         | $0.502{\pm}0.014$   |
| 90% RH    | 0.156±0.010       | 0.183±0.013       | 0.189±0.013         | $0.492 \pm 0.017$ | $0.580 \pm 0.018$   | $0.506 \pm 0.016$   |
| SOA Yield | 0.379±0.039       | 0.337±0.048       | 0.288±0.038         | 0.511±0.097       | $0.479 \pm 0.044$   | $0.401 \pm 0.068$   |

**Table 3.** The SOA yield and mass fraction of particulate peroxides at low or high [O3]/[limonene] ratio in the presence or absence of OH scavenger from 0% to 90% relative humidity (RH).

Note: L, low [O3]/[limonene]ratio; H, high [O3]/[limonene]ratio; No-sca, none scavenger; 2-But, 2-butanol; C-hex,

cyclohexane.

**Table 4.** Yields (%) of carbonyls at low or high [O3]/[limonene] ratio in the presence or absence of OH scavenger.

|           | HACE            | FA                              | AA        | ACE             | GL              | MGL       |
|-----------|-----------------|---------------------------------|-----------|-----------------|-----------------|-----------|
| L(No-sca) | $2.04 \pm 0.48$ | 7.02±0.90-10.58±0.94            | 1.32±0.24 | 0.22±0.15       | 0.89±0.25       | 0.56±0.34 |
| L(2-But)  | 3.94±0.55       | $4.90 \pm 0.86 - 7.77 \pm 0.86$ | 3.98±0.60 | 0.35±0.18       | $0.69{\pm}0.25$ | 0.55±0.44 |
| L(C-hex)  | —               | 5.21±0.66-8.25±0.55             | —         | —               | $0.81 \pm 0.24$ | 0.67±0.56 |
| H(No-sca) | 4.45±0.52       | 13.11±0.63-27.00±1.56           | 2.16±0.84 | 0.79±0.22       | 1.33±0.41       | 1.35±0.61 |
| H(2-But)  | 10.15±2.11      | 11.03±0.77-23.33±0.62           | 7.86±1.32 | $1.00{\pm}0.28$ | 1.25±0.36       | 1.31±0.21 |
| H(C-hex)  | —               | 10.80±1.28-23.32±1.21           | 1.89±1.22 | $0.60{\pm}0.47$ | 1.25±0.51       | 1.23±0.22 |

Note: —, below detection limit; L, low [O3]/[limonene]ratio; H, high [O3]/[limonene]ratio; No-sca, none scavenger; 2-But,

2-butanol; C-hex, cyclohexane; HACE, hydroxyacetone; FA, formaldehyde; AA, acetaldehyde; ACE, acetone; GL, glyoxal;

MGL, methylglyoxal.